# Communication Efficient Distributed Learning for Kernelized Contextual Bandits

**Chuanhao Li**[1]   **Huazheng Wang**[2]   **Mengdi Wang**[3]   **Hongning Wang**[1]
[1]University of Virginia   [2]Oregon State University   [3]Princeton University
{cl5ev,hw5x}@virginia.edu
huazheng.wang@oregonstate.edu   mengdiw@princeton.edu

## Abstract

We tackle the communication efficiency challenge of learning kernelized contextual bandits in a distributed setting. Despite the recent advances in communication-efficient distributed bandit learning, existing solutions are restricted to simple models like multi-armed bandits and linear bandits, which hamper their practical utility. In this paper, instead of assuming the existence of a linear reward mapping from the features to the expected rewards, we consider non-linear reward mappings, by letting agents collaboratively search in a reproducing kernel Hilbert space (RKHS). This introduces significant challenges in communication efficiency as distributed kernel learning requires the transfer of raw data, leading to a communication cost that grows linearly w.r.t. time horizon $T$. We address this issue by equipping all agents to communicate via a common Nyström embedding that gets updated adaptively as more data points are collected. We rigorously proved that our algorithm can attain sub-linear rate in both regret and communication cost.

## 1 Introduction

Contextual bandit algorithms have been widely used for a variety of real-world applications, including recommender systems [20], display advertisement [21] and clinical trials [11]. While most existing bandit solutions assume a centralized setting (i.e., all the data reside in and all the actions are taken by a central server), there is increasing research effort on distributed bandit learning lately [30, 10, 24, 17, 19], where $N$ clients, under the coordination of a central server, collaborate to minimize the overall cumulative regret incurred over a finite time horizon $T$. In many distributed application scenarios, communication is the main bottleneck, e.g., communication in a network of mobile devices can be slower than local computation by several orders of magnitude [16]. Therefore, it is vital for distributed bandit learning algorithms to attain sub-linear rate (w.r.t. time horizon $T$) in both cumulative regret and communication cost.

However, prior works in this line of research are restricted to linear models [30], which could oversimplify the problem and thus leads to inferior performance in practice. In centralized setting, kernelized bandit algorithms, e.g., KernelUCB [29] and IGP-UCB [6], are proposed to address this issue by modeling the unknown reward mapping as a non-parametric function lying in a reproducing kernel Hilbert space (RKHS), i.e., the expected reward is linear w.r.t. an action feature map of possibly infinite dimensions. Despite the strong modeling capability of kernel method, collaborative exploration in the RKHS gives rise to additional challenges in designing a communication efficient bandit algorithm. Specifically, unlike distributed linear bandit where the clients can simply communicate the $d \times d$ sufficient statistics [30], where $d$ is the dimension of the input feature vector, the *joint kernelized estimation* of the unknown reward function requires communicating either 1) the $p \times p$ sufficient statistics in the RKHS, where $p$ is the dimension of the RKHS that is possibly infinite, or 2) the set of input feature vectors that grows linearly w.r.t. $T$. Neither of them is practical due to the huge communication cost.

36th Conference on Neural Information Processing Systems (NeurIPS 2022).

In this paper, we propose the first communication efficient algorithm for distributed kernel bandits, which tackles the aforementioned challenge via a low-rank approximation of the empirical kernel matrix. In particular, we extended the Nyström method [22] to distributed learning for kernelized contextual bandits. In this solution, all clients first project their local data to a finite RKHS spanned by a common dictionary, i.e., a small subset of the original dataset, and then they only need to communicate the embedded statistics for collaborative exploration. To ensure effective regret reduction after each communication round, as well as ensuring the dictionary remains representative for the entire distributed dataset throughout the learning process, the frequency of dictionary update and synchronization of embedded statistics is controlled by measuring the amount of new information each client has gained since last communication. We rigorously prove that the proposed algorithm incurs an $O(N^2 \gamma_{NT}^3)$ communication cost, where $\gamma_{NT}$ is the maximum information gain that is known to be $O\big(\log(NT)\big)$ for kernels with exponentially decaying eigenvalues, which includes the most commonly used Gaussian kernel, while attaining the optimal $O(\sqrt{NT}\gamma_{NT})$ cumulative regret.

## 2   Related Works

To balance exploration and exploitation in stochastic linear contextual bandits, LinUCB algorithm [20, 1] is commonly used, which selects arm optimistically w.r.t. a constructed confidence set on the unknown linear reward function. By using kernels and Gaussian processes, studies in [26, 29, 6] further extend UCB algorithms to non-parametric reward functions in RKHS, i.e., the feature map associated with each arm is possibly infinite.

Recent years have witnessed increasing research efforts in distributed bandit learning, i.e., multiple agents collaborate in pure exploration [15, 27, 7], or regret minimization [24, 30, 19]. They mainly differ in the relations of learning problems solved by the agents (i.e., homogeneous vs., heterogeneous) and the type of communication network (i.e., peer-to-peer (P2P) vs., star-shaped). Most of these works assume linear reward functions, and the clients communicate by transferring the $O(d^2)$ sufficient statistics. Korda et al. [18] considered a peer-to-peer (P2P) communication network and assumed that the clients form clusters, i.e., each cluster is associated with a unique bandit problem. Huang et al. [17] considered a star-shaped communication network as in our paper, but their proposed phase-based elimination algorithm only works in the fixed arm set setting. The closest works to ours are [30, 10, 19], which proposed event-triggered communication protocols to obtain sub-linear communication cost over time for distributed linear bandits with a time-varying arm set. In comparison, distributed kernelized contextual bandits still remain under-explored. The only existing work in this direction [9] considered heterogeneous agents, where each agent is associated with an additional feature describing the task similarity between agents. However, they assumed a local communication setting, where the agent immediately shares the new raw data point to its neighbors after each interaction, and thus the communication cost is still linear over time.

Another closely related line of works is kernelized bandits with approximation, where Nyström method is adopted to improve computation efficiency in a centralized setting. Calandriello et al. [3] proposed an algorithm named BKB, which uses Ridge Leverage Score sampling (RLS) to re-sample a new dictionary from the updated dataset after each interaction with the environment. A recent work by Zenati et al. [31] further improved the computation efficiency of BKB by adopting an online sampling method to update the dictionary. However, both of them updated the dictionary at each time step to ensure the dictionary remains representative w.r.t. the growing dataset, and therefore are not applicable to our problem. This is because the dataset is stored cross clients in a distributed manner, and projecting the dataset to the space spanned by the new dictionary requires communication with all clients, which is prohibitively expensive in terms of communication. Calandriello et al. [4] also proposed a variant of BKB, named BBKB, for batched Gaussian process optimization. BBKB only needs to update the dictionary occasionally according to an adaptive schedule, and thus partially addresses the issue mentioned above. However, as BBKB works in a centralized setting, their adaptive schedule can be computed based on the whole batch of data, while in our decentralized setting, each client can only make the update decision according to the data that is locally available. Moreover, in BBKB, all the interactions are based on a fixed model estimation over the whole batch, which is mentioned in their Appendix A.4 as a result of an inherent technical difficulty. In comparison, our proposed method effectively addresses this difficulty with improved analysis, and thus allows each client to utilize newly collected data to update its model estimation on the fly.

# 3 Preliminaries

In this section, we first formulate the problem of distributed kernelized contextual bandits. Then, as a starting point, we propose and analyze a naive UCB-type algorithm for distributed kernelized contextual bandit problem, named DisKernelUCB. This demonstrates the challenges in designing a communication efficient algorithm for this problem, and also lays down the foundation for further improvement on communication efficiency in Section 4.

## 3.1 Distributed Kernelized Contextual Bandit Problem

Consider a learning system with 1) $N$ clients that are responsible for taking actions and receiving feedback from the environment, and 2) a central server that coordinates the communication among the clients. The clients cannot directly communicate with each other, but only with the central server, i.e., a star-shaped communication network. Following prior works [30, 10], we assume the $N$ clients interact with the environment in a round-robin manner for a total number of $T$ rounds.

Specifically, at round $l \in [T]$, each client $i \in [N]$ chooses an arm $\mathbf{x}_t$ from a candidate set $\mathcal{A}_t$, and then receives the corresponding reward feedback $y_t = f(\mathbf{x}_t) + \eta_t \in \mathbb{R}$, where the subscript $t := N(l-1)+i$ indicates this is the $t$-th interaction between the learning system and the environment, and we refer to it as time step $t$ [1]. Note that $\mathcal{A}_t$ is a time-varying subset of $\mathcal{A} \subseteq \mathbb{R}^d$ that is possibly infinite, $f$ denotes the unknown reward function shared by all the clients, and $\eta_t$ denotes the noise.

Denote the sequence of indices corresponding to the interactions between client $i$ and the environment up to time $t$ as $\mathcal{N}_t(i) = \{1 \le s \le t : i_s = i\}$ (if $s \bmod N = 0$, then $i_s = N$; otherwise $i_s = s \bmod N$) for $t = 1, 2, \ldots, NT$. By definition, $|\mathcal{N}_{Nl}(i)| = l, \forall l \in [T]$, i.e., the clients have equal number of interactions at the end of each round $l$.

**Kernelized Reward Function**   We consider an unknown reward function $f$ that lies in a RKHS, denoted as $\mathcal{H}$, such that the reward can be equivalently written as

$$y_t = \theta_\star^\top \phi(\mathbf{x}_t) + \eta_t,$$

where $\theta_\star \in \mathcal{H}$ is an unknown parameter, and $\phi : \mathbb{R}^d \to \mathcal{H}$ is a known feature map associated with $\mathcal{H}$. We assume $\eta_t$ is zero-mean $R$-sub-Gaussian conditioned on $\sigma\big((\mathbf{x}_s, \eta_s)_{s \in \mathcal{N}_{t-1}(i_t)}\big), \forall t$, which denotes the $\sigma$-algebra generated by client $i_t$'s previously pulled arms and the corresponding noise. In addition, there exists a positive definite kernel $k(\cdot, \cdot)$ associated with $\mathcal{H}$, and we assume $\forall \mathbf{x} \in \mathcal{A}$ that, $\|\mathbf{x}\|_k \le L$ and $\|f\|_k \le S$ for some $L, S > 0$.

**Regret and Communication Cost**   The goal of the learning system is to minimize the cumulative (pseudo) regret for all $N$ clients, i.e., $R_{NT} = \sum_{t=1}^{NT} r_t$, where $r_t = \max_{\mathbf{x} \in \mathcal{A}_t} \phi(\mathbf{x})^\top \theta_\star - \phi(\mathbf{x}_t)^\top \theta_\star$. Meanwhile, the learning system also wants to keep the communication cost $C_{NT}$ low, which is measured by the total number of scalars being transferred across the system up to time step $NT$.

## 3.2 Distributed Kernel UCB

As a starting point to studying the communication efficient algorithm in Section 4 and demonstrate the challenges in designing a communication efficient distributed kernelized contextual bandit algorithm, here we first introduce and analyze a naive algorithm where the $N$ clients collaborate on learning the exact parameters of kernel bandit, i.e., the mean and variance of estimated reward. We name this algorithm Distributed Kernel UCB, or DisKernelUCB for short, and its description is given in Algorithm 1.

**Arm Selection**   For each round $l \in [T]$, when client $i \in [N]$ interacts with the environment, i.e., the $t$-th interaction between the learning system and the environment where $t = N(l-1)+i$, it chooses arm $\mathbf{x}_t \in \mathcal{A}_t$ based on the UCB of the mean estimator (line 5):

$$\mathbf{x}_t = \arg\max_{\mathbf{x} \in \mathcal{A}_t} \hat{\mu}_{t-1,i}(\mathbf{x}) + \alpha_{t-1,i} \hat{\sigma}_{t-1,i}(\mathbf{x}) \tag{1}$$

---

[1]The meaning of index $t$ is slightly different from prior works, e.g. DisLinUCB in [30], but this is only to simplify the use of notation and does not affect the theoretical results

---

**Algorithm 1** Distributed Kernel UCB (DisKernelUCB)

---

1: **Input** threshold $D > 0$
2: **Initialize** $t_{\text{last}} = 0$, $\mathcal{D}_0(i) = \Delta\mathcal{D}_0(i) = \emptyset, \forall i \in [N]$
3: **for** round $l = 1, 2, ..., T$ **do**
4:     **for** client $i = 1, 2, ..., N$ **do**
5:         Client $i$ chooses arm $\mathbf{x}_t \in \mathcal{A}_t$ according to Eq (1) and observes reward $y_t$, where $t = N(l-1) + i$
6:         Client $i$ updates $\mathbf{K}_{\mathcal{D}_t(i),\mathcal{D}_t(i)}, \mathbf{y}_{\mathcal{D}_t(i)}$, where $\mathcal{D}_t(i) = \mathcal{D}_{t-1}(i) \cup \{t\}$; and its upload buffer $\Delta\mathcal{D}_t(i) = \Delta\mathcal{D}_{t-1}(i) \cup \{t\}$
        *// Global Synchronization*
7:         **if** the event $\mathcal{U}_t(D)$ defined in Eq (2) is true **then**
8:             **Clients** $\forall j \in [N]$: send $\{(\mathbf{x}_s, y_s)\}_{s \in \Delta\mathcal{D}_t(j)}$ to server, and reset $\Delta\mathcal{D}_t(j) = \emptyset$
9:             **Server**: aggregates and sends back $\{(\mathbf{x}_s, y_s)\}_{s \in [t]}$; sets $t_{\text{last}} = t$
10:           **Clients** $\forall j \in [N]$: update $\mathbf{K}_{\mathcal{D}_t(j),\mathcal{D}_t(j)}, \mathbf{y}_{\mathcal{D}_t(i)}$, where $\mathcal{D}_t(j) = [t]$

---

where $\hat{\mu}_{t,i}(\mathbf{x})$ and $\hat{\sigma}_{t,i}^2(\mathbf{x})$ denote client $i$'s local estimated mean reward for arm $\mathbf{x} \in \mathcal{A}$ and its variance, and $\alpha_{t-1,i}$ is a carefully chosen scaling factor to balance exploration and exploitation (see Lemma 3.1 for proper choice).

To facilitate further discussion, for time step $t \in [NT]$, we denote the sequence of time indices for the data points that have been used to update client $i$'s local estimate as $\mathcal{D}_t(i)$, which include both data points collected locally and those shared by the other clients. If the clients never communicate, $\mathcal{D}_t(i) = \mathcal{N}_t(i), \forall t, i$; otherwise, $\mathcal{N}_t(i) \subset \mathcal{D}_t(i) \subseteq [t]$, with $\mathcal{D}_t(i) = [t]$ recovering the centralized setting, i.e., each new data point collected from the environment immediately becomes available to all the clients in the learning system. The design matrix and reward vector for client $i$ at time step $t$ are denoted by $\mathbf{X}_{\mathcal{D}_t(i)} = [\mathbf{x}_s]_{s \in \mathcal{D}_t(i)}^\top \in \mathbb{R}^{|\mathcal{D}_t(i)| \times d}, \mathbf{y}_{t,i} = [y_s]_{s \in \mathcal{D}_t(i)}^\top \in \mathbb{R}^{|\mathcal{D}_t(i)|}$, respectively. By applying the feature map $\phi(\cdot)$ to each row of $\mathbf{X}_{\mathcal{D}_t(i)}$, we obtain $\mathbf{\Phi}_{\mathcal{D}_t(i)} \in \mathbb{R}^{|\mathcal{D}_t(i)| \times p}$, where $p$ is the dimension of $\mathcal{H}$ and is possibly infinite. Since the reward function is linear in $\mathcal{H}$, client $i$ can construct the Ridge regression estimator $\hat{\theta}_{t,i} = (\mathbf{\Phi}_{\mathcal{D}_t(i)}^\top \mathbf{\Phi}_{\mathcal{D}_t(i)} + \lambda I)^{-1} \mathbf{\Phi}_{\mathcal{D}_t(i)}^\top \mathbf{y}_{t,i}$, where $\lambda > 0$ is the regularization coefficient. This gives us the estimated mean reward and variance in primal form for any arm $\mathbf{x} \in \mathcal{A}$, i.e., $\hat{\mu}_{t,i}(\mathbf{x}) = \phi(\mathbf{x})^\top \mathbf{A}_{t,i}^{-1} \mathbf{b}_{t,i}$ and $\hat{\sigma}_{t,i}(\mathbf{x}) = \sqrt{\phi(\mathbf{x})^\top \mathbf{A}_{t,i}^{-1} \phi(\mathbf{x})}$, where $\mathbf{A}_{t,i} = \mathbf{\Phi}_{\mathcal{D}_t(i)}^\top \mathbf{\Phi}_{\mathcal{D}_t(i)} + \lambda \mathbf{I}$ and $\mathbf{b}_{t,i} = \mathbf{\Phi}_{\mathcal{D}_t(i)}^\top \mathbf{y}_{t,i}$. Then using the kernel trick, we can obtain their equivalence in the dual form that only involves entries of the kernel matrix, and avoids directly working on $\mathcal{H}$ which is possibly infinite:

$$\hat{\mu}_{t,i}(\mathbf{x}) = \mathbf{K}_{\mathcal{D}_t(i)}(\mathbf{x})^\top \big(\mathbf{K}_{\mathcal{D}_t(i),\mathcal{D}_t(i)} + \lambda I\big)^{-1} \mathbf{y}_{\mathcal{D}_t(i)}$$

$$\hat{\sigma}_{t,i}(\mathbf{x}) = \lambda^{-1/2} \sqrt{k(\mathbf{x},\mathbf{x}) - \mathbf{K}_{\mathcal{D}_t(i)}(\mathbf{x})^\top \big(\mathbf{K}_{\mathcal{D}_t(i),\mathcal{D}_t(i)} + \lambda I\big)^{-1} \mathbf{K}_{\mathcal{D}_t(i)}(\mathbf{x})}$$

where $\mathbf{K}_{\mathcal{D}_t(i)}(\mathbf{x}) = \mathbf{\Phi}_{\mathcal{D}_t(i)} \phi(\mathbf{x}) = [k(\mathbf{x}_s, \mathbf{x})]_{s \in \mathcal{D}_t(i)}^\top \in \mathbb{R}^{|\mathcal{D}_t(i)|}$, and $\mathbf{K}_{\mathcal{D}_t(i),\mathcal{D}_t(i)} = \mathbf{\Phi}_{\mathcal{D}_t(i)}^\top \mathbf{\Phi}_{\mathcal{D}_t(i)} = [k(\mathbf{x}_s, \mathbf{x}_{s'})]_{s,s' \in \mathcal{D}_t(i)} \in \mathbb{R}^{|\mathcal{D}_t(i)| \times |\mathcal{D}_t(i)|}$.

**Communication Protocol** To reduce the regret in future interactions with the environment, the $N$ clients need to collaborate via communication, and a carefully designed communication protocol is essential in ensuring the communication efficiency. In prior works like DisLinUCB [30], after each round of interaction with the environment, client $i$ checks whether the event $\{(|\mathcal{D}_t(i)| - |\mathcal{D}_{t_{\text{last}}}(i)|) \log(\frac{\det(\mathbf{A}_{t,i})}{\det(\mathbf{A}_{t_{\text{last}},i})}) > D\}$ is true, where $t_{\text{last}}$ denotes the time step of last global synchronization. If true, a new global synchronization is triggered, such that the server will require all clients to upload their sufficient statistics since $t_{\text{last}}$, aggregate them to compute $\{\mathbf{A}_t, \mathbf{b}_t\}$, and then synchronize the aggregated sufficient statistics with all clients, i.e., set $\{\mathbf{A}_{t,i}, \mathbf{b}_{t,i}\} = \{\mathbf{A}_t, \mathbf{b}_t\}, \forall i \in [N]$.

Using kernel trick, we can obtain an equivalent event-trigger in terms of the kernel matrix,

$$\mathcal{U}_t(D) = \left\{ (|\mathcal{D}_t(i_t)| - |\mathcal{D}_{t_{\text{last}}}(i_t)|) \log \left( \frac{\det(\mathbf{I} + \lambda^{-1} \mathbf{K}_{\mathcal{D}_t(i_t),\mathcal{D}_t(i_t)})}{\det(\mathbf{I} + \lambda^{-1} \mathbf{K}_{\mathcal{D}_t(i_t) \setminus \Delta\mathcal{D}_t(i_t), \mathcal{D}_t(i_t) \setminus \Delta\mathcal{D}_t(i_t)})} \right) > D \right\}.$$
(2)

where $D > 0$ denotes the predefined threshold value. If event $\mathcal{U}_t(D)$ is true (line 7), a global synchronization is triggered (line 7-10), where the local datasets of all $N$ clients are synchronized to $\{(\mathbf{x}_s, y_s)\}_{s \in [t]}$. We should note that the transfer of raw data $(\mathbf{x}_s, y_s)$ is necessary for the update of the kernel matrix and reward vector in line 6 and line 10, which will be used for arm selection at line 5. This is an inherent disadvantage of kernelized estimation in distributed settings, which, as we mentioned in Section 2, is also true for the existing distributed kernelized bandit algorithm [9]. Lemma 3.1 below shows that in order to obtain the optimal order of regret, DisKernelUCB incurs a communication cost linear in $T$ (proof given in the appendix), which is expensive for an online learning problem.

**Lemma 3.1** (Regret and Communication Cost of DisKernelUCB). *With threshold $D = \frac{T}{N \gamma_{NT}}$,* $\alpha_{t,i} = \sqrt{\lambda}\|\theta_\star\| + R\sqrt{4 \ln NT/\delta + 2 \ln \det(\mathbf{I} + \mathbf{K}_{\mathcal{D}_t(i), \mathcal{D}_t(i)}/\lambda)}$, *we have*

$$R_{NT} = O\big(\sqrt{NT}(\|\theta_\star\|\sqrt{\gamma_{NT}} + \gamma_{NT})\big),$$

*with probability at least $1 - \delta$, and*

$$C_{NT} = O(TN^2 d).$$

*where $\gamma_{NT} := \max_{\mathcal{D} \subset \mathcal{A}: |\mathcal{D}| = NT} \frac{1}{2} \log \det(\mathbf{K}_{\mathcal{D}, \mathcal{D}}/\lambda + \mathbf{I})$ is the maximum information gain after $NT$ interactions [6]. It is problem-dependent and can be bounded for specific arm set $\mathcal{A}$ and kernel function $k(\cdot, \cdot)$. For example, $\gamma_{NT} = O(d \log(NT))$ for linear kernel and $\gamma_{NT} = O(\log(NT)^{d+1})$ for Gaussian kernel.*

**Remark 1.** *In the distributed linear bandit problem, to attain $O(d\sqrt{NT} \ln(NT))$ regret, DisLinUCB [30] requires a total number of $O(N^{0.5} d \log(NT))$ synchronizations, and DisKernelUCB matches this result under linear kernel, as it requires $O(N^{0.5} \gamma_{NT})$ synchronizations. We should note that the communication cost for each synchronization in DisLinUCB is fixed, i.e., $O(Nd^2)$ to synchronize the sufficient statistics with all the clients, so in total $C_{NT} = O(N^{1.5} d^3 \ln(NT))$. However, this is not the case for DisKernelUCB that needs to send raw data, because the communication cost for each synchronization in DisKernelUCB is not fixed, but depends on the number of unshared data points on each client. Even if the total number of synchronizations is small, DisKernelUCB could still incur $C_{NT} = O(TN^2 d)$ in the worse case. Consider the extreme case where synchronization only happens once, but it happens near $NT$, then we still have $C_{NT} = O(TN^2 d)$. The time when synchronization gets triggered depends on $\{\mathcal{A}_t\}_{t \in [NT]}$, which is out of the control of the algorithm. Therefore, in the following section, to improve the communication efficiency of DisKernelUCB, we propose to let each client communicate embedded statistics in some small subspace during each global synchronization.*

## 4 Approximated Distributed Kernel UCB

In this section, we propose and analyze a new algorithm that improves the communication efficiency of DisKernelUCB using the Nyström approximation, such that the clients only communicate the embedded statistics during event-triggered synchronizations. We name this algorithm Approximated Distributed Kernel UCB, or Approx-DisKernelUCB for short. Its description is given in Algorithm 2.

### 4.1 Algorithm

**Arm selection** For each round $l \in [T]$, when client $i \in [N]$ interacts with the environment, i.e., the $t$-th interaction between the learning system and the environment where $t := N(l-1) + i$, instead of using the UCB for the exact estimator in Eq (1), client $i$ chooses arm $\mathbf{x}_t \in \mathcal{A}_t$ that maximizes the UCB for the approximated estimator (line 5):

$$\mathbf{x}_t = \arg\max_{\mathbf{x} \in \mathcal{A}_{t,i}} \tilde{\mu}_{t-1,i}(\mathbf{x}) + \alpha_{t-1,i} \tilde{\sigma}_{t-1,i}(\mathbf{x}) \tag{3}$$

where $\tilde{\mu}_{t-1,i}(\mathbf{x})$ and $\tilde{\sigma}_{t-1,i}(\mathbf{x})$ are approximated using Nyeström method, and the statistics used to compute these approximations are much more efficient to communicate as they scale with the maximum information gain $\gamma_{NT}$ instead of $T$.

Specifically, Nyström method works by projecting some original dataset $\mathcal{D}$ to the subspace defined by a small representative subset $\mathcal{S} \subseteq \mathcal{D}$, which is called the dictionary. The orthogonal projection matrix is defined as

$$\mathbf{P}_{\mathcal{S}} = \Phi_{\mathcal{S}}^\top \big(\Phi_{\mathcal{S}} \Phi_{\mathcal{S}}^\top\big)^{-1} \Phi_{\mathcal{S}} = \Phi_{\mathcal{S}}^\top \mathbf{K}_{\mathcal{S}, \mathcal{S}}^{-1} \Phi_{\mathcal{S}} \in \mathbb{R}^{p \times p}$$

---

**Algorithm 2** Approximated Distributed Kernel UCB (Approx-DisKernelUCB)

---

1: **Input:** threshold $D > 0$, regularization parameter $\lambda > 0$, $\delta \in (0, 1)$ and kernel function $k(\cdot, \cdot)$.
2: **Initialize** $\tilde{\mu}_{0,i}(\mathbf{x}) = 0, \tilde{\sigma}_{0,i}(\mathbf{x}) = \sqrt{k(\mathbf{x}, \mathbf{x})}, \mathcal{N}_0(i) = \mathcal{D}_0(i) = \emptyset, \forall i \in [N]; \mathcal{S}_0 = \emptyset, t_{\text{last}} = 0$
3: **for** round $l = 1, 2, ..., T$ **do**
4:     **for** client $i = 1, 2, ..., N$ **do**
5:         [Client $i$] selects arm $\mathbf{x}_t \in \mathcal{A}_t$ according to Eq (3) and observes reward $y_t$, where $t := N(l-1) + i$
6:         [Client $i$] updates $\mathbf{Z}_{\mathcal{D}_t(i);\mathcal{S}_{t_{\text{last}}}}^\top \mathbf{Z}_{\mathcal{D}_t(i);\mathcal{S}_{t_{\text{last}}}}$ and $\mathbf{Z}_{\mathcal{D}_t(i);\mathcal{S}_{t_{\text{last}}}}^\top \mathbf{y}_{\mathcal{D}_t(i)}$ using $(\mathbf{z}(\mathbf{x}_t; \mathcal{S}_{t_{\text{last}}}), y_t)$; sets
           $\mathcal{N}_t(i) = \mathcal{N}_{t-1}(i) \cup \{t\}$, and $\mathcal{D}_t(i) = \mathcal{D}_{t-1}(i) \cup \{t\}$
           *// Global Synchronization*
7:         **if** the event $\mathcal{U}_t(D)$ defined in Eq (4) is true **then**
8:             [Clients $\forall i$] sample $\mathcal{S}_{t,i} = \text{RLS}(\mathcal{N}_t(i), \bar{q}, \tilde{\sigma}_{t_{\text{last}},i}^2)$, and send $\{(\mathbf{x}_s, y_s)\}_{s \in \mathcal{S}_{t,i}}$ to server
9:             [Server] aggregates and sends $\{(\mathbf{x}_s, y_s)\}_{s \in \mathcal{S}_t}$ back to all clients, where $\mathcal{S}_t = \cup_{i \in [N]} \mathcal{S}_{t,i}$
10:           [Clients $\forall i$] compute and send $\{\mathbf{Z}_{\mathcal{N}_t(i);\mathcal{S}_t}^\top \mathbf{Z}_{\mathcal{N}_t(i);\mathcal{S}_t}, \mathbf{Z}_{\mathcal{N}_t(i);\mathcal{S}_t}^\top \mathbf{y}_{\mathcal{N}_t(i)}\}$ to server
11:           [Server] aggregates $\sum_{i=1}^N \mathbf{Z}_{\mathcal{N}_t(i);\mathcal{S}_t}^\top \mathbf{Z}_{\mathcal{N}_t(i);\mathcal{S}_t}, \sum_{i=1}^N \mathbf{Z}_{\mathcal{N}_t(i);\mathcal{S}_t}^\top \mathbf{y}_{\mathcal{N}_t(i)}$ and sends it back
12:           [Clients $\forall i$] updates $\mathbf{Z}_{\mathcal{D}_t(i);\mathcal{S}_t}^\top \mathbf{Z}_{\mathcal{D}_t(i);\mathcal{S}_t}$ and $\mathbf{Z}_{\mathcal{D}_t(i);\mathcal{S}_t}^\top \mathbf{y}_{\mathcal{D}_t(i)}$; sets $\mathcal{D}_t(i) = \cup_{i=1}^N \mathcal{N}_t(i) = [t]$ and $t_{\text{last}} = t$

---

We then take eigen-decomposition of $\mathbf{K}_{\mathcal{S},\mathcal{S}} = \mathbf{U}\mathbf{\Lambda}\mathbf{U}^\top$ to rewrite the orthogonal projection as $\mathbf{P}_{\mathcal{S}} = \mathbf{\Phi}_{\mathcal{S}}^\top \mathbf{U}\mathbf{\Lambda}^{-1/2}\mathbf{\Lambda}^{-1/2}\mathbf{U}^\top \mathbf{\Phi}_{\mathcal{S}}$, and define the Nyström embedding function

$$z(\mathbf{x}; \mathcal{S}) = \mathbf{P}_{\mathcal{S}}^{1/2}\phi(\mathbf{x}) = \mathbf{\Lambda}^{-1/2}\mathbf{U}^\top \mathbf{\Phi}_{\mathcal{S}}\phi(\mathbf{x}) = \mathbf{K}_{\mathcal{S},\mathcal{S}}^{-1/2}\mathbf{K}_{\mathcal{S}}(\mathbf{x})$$

which maps the data point $\mathbf{x}$ from $\mathbb{R}^d$ to $\mathbb{R}^{|\mathcal{S}|}$.

Therefore, we can approximate the Ridge regression estimator in Section 3.2 as $\tilde{\theta}_{t,i} = \tilde{\mathbf{A}}_{t,i}^{-1}\tilde{\mathbf{b}}_{t,i}$, where $\tilde{\mathbf{A}}_{t,i} = \mathbf{P}_{\mathcal{S}}\mathbf{\Phi}_{\mathcal{D}_t(i)}^\top \mathbf{\Phi}_{\mathcal{D}_t(i)}\mathbf{P}_{\mathcal{S}} + \lambda\mathbf{I}$, and $\tilde{\mathbf{b}}_{t,i} = \mathbf{P}_{\mathcal{S}}\mathbf{\Phi}_{\mathcal{D}_t(i)}^\top \mathbf{y}_{\mathcal{D}_t(i)}$, and thus the approximated mean reward and variance in Eq (3) can be expressed as $\tilde{\mu}_{t,i}(\mathbf{x}) = \phi(\mathbf{x})^\top \tilde{\mathbf{A}}_{t,i}^{-1}\tilde{\mathbf{b}}_{t,i}$ and $\tilde{\sigma}_{t,i}(\mathbf{x}) = \sqrt{\phi(\mathbf{x})^\top \tilde{\mathbf{A}}_{t,i}^{-1}\phi(\mathbf{x})}$, and their kernelized representation are (see appendix for detailed derivation)

$$\tilde{\mu}_{t,i}(\mathbf{x}) = z(\mathbf{x}; \mathcal{S})^\top \left(\mathbf{Z}_{\mathcal{D}_t(i);\mathcal{S}}^\top \mathbf{Z}_{\mathcal{D}_t(i);\mathcal{S}} + \lambda\mathbf{I}\right)^{-1}\mathbf{Z}_{\mathcal{D}_t(i);\mathcal{S}}^\top \mathbf{y}_{\mathcal{D}_t(i)}$$

$$\tilde{\sigma}_{t,i}(\mathbf{x}) = \lambda^{-1/2}\sqrt{k(\mathbf{x}, \mathbf{x}) - z(\mathbf{x}; \mathcal{S})^\top \mathbf{Z}_{\mathcal{D}_t(i);\mathcal{S}}^\top \mathbf{Z}_{\mathcal{D}_t(i);\mathcal{S}}[\mathbf{Z}_{\mathcal{D}_t(i);\mathcal{S}}^\top \mathbf{Z}_{\mathcal{D}_t(i);\mathcal{S}} + \lambda\mathbf{I}]^{-1}z(\mathbf{x}|\mathcal{S})}$$

where $\mathbf{Z}_{\mathcal{D}_t(i);\mathcal{S}} \in \mathbb{R}^{|\mathcal{D}_t(i)| \times |\mathcal{S}|}$ is obtained by applying $z(\cdot; \mathcal{S})$ to each row of $\mathbf{X}_{\mathcal{D}_t(i)}$, i.e., $\mathbf{Z}_{\mathcal{D}_t(i);\mathcal{S}} = \mathbf{\Phi}_{\mathcal{D}_t(i)}\mathbf{P}_{\mathcal{S}}^{1/2}$. We can see that the computation of $\tilde{\mu}_{t,i}(\mathbf{x})$ and $\tilde{\sigma}_{t,i}(\mathbf{x})$ only requires the embedded statistics: matrix $\mathbf{Z}_{\mathcal{D}_t(i);\mathcal{S}}^\top \mathbf{Z}_{\mathcal{D}_t(i);\mathcal{S}} \in \mathbb{R}^{|\mathcal{S}| \times |\mathcal{S}|}$ and vector $\mathbf{Z}_{\mathcal{D}_t(i);\mathcal{S}}^\top \mathbf{y}_{\mathcal{D}_t(i)} \in \mathbb{R}^{|\mathcal{S}|}$, which, as we will show later, makes joint kernelized estimation among $N$ clients much more efficient in communication.

After obtaining the new data point $(\mathbf{x}_t, y_t)$, client $i$ immediately updates both $\tilde{\mu}_{t-1,i}(\mathbf{x})$ and $\tilde{\sigma}_{t-1,i}(\mathbf{x})$ using the newly collected data point $(\mathbf{x}_t, y_t)$, i.e., by projecting $\mathbf{x}_t$ to the finite dimensional RKHS spanned by $\mathbf{\Phi}_{\mathcal{S}_{t_{\text{last}}}}$ (line 6). Recall that, we use $\mathcal{N}_t(i)$ to denote the sequence of indices for data collected by client $i$, and denote by $\mathcal{D}_t(i)$ the sequence of indices for data that has been used to update client $i$'s model estimation $\tilde{\mu}_{t,i}$. Therefore, both of them need to be updated to include time step $t$.

**Communication Protocol** With the approximated estimator, the size of message being communicated across the learning system is reduced. However, a carefully designed event-trigger is still required to minimize the total number of global synchronizations up to time $NT$. Since the clients can no longer evaluate the exact kernel matrices in Eq (2), we instead use the event-trigger in Eq (4), which can be computed using the approximated variance from last global synchronization as,

$$\mathcal{U}_t(D) = \left\{ \sum_{s \in \mathcal{D}_t(i) \backslash \mathcal{D}_{t_{\text{last}}}(i)} \tilde{\sigma}_{t_{\text{last}},i}^2(\mathbf{x}_s) > D \right\} \tag{4}$$

Similar to Algorithm 1, if Eq (4) is true, global synchronization is triggered, where both the dictionary and the embedded statistics get updated. During synchronization, each client first samples a subset $\mathcal{S}_t(i)$ from $\mathcal{N}_t(i)$ (line 8) using Ridge Leverage Score sampling (RLS) [3, 4], which is given in Algorithm 3, and then sends $\{(\mathbf{x}_s, y_s)\}_{s \in \mathcal{S}_t(i)}$ to the server. The server aggregates the received local subsets to construct a new dictionary $\{(\mathbf{x}_s, y_s)\}_{s \in \mathcal{S}_t}$, where $\mathcal{S}_t = \cup_{i=1}^N \mathcal{S}_t(i)$, and then sends it back to all $N$ clients (line 9). Finally, the $N$ clients use this updated dictionary to re-compute the embedded statistics of their local data, and then synchronize it with all other clients via the server (line 10-12).

---

**Algorithm 3**  Ridge Leverage Score Sampling (RLS)

1: **Input:** dataset $\mathcal{D}$, scaling factor $\bar{q}$, (possibly delayed and approximated) variance function $\tilde{\sigma}^2(\cdot)$
2: **Initialize** new dictionary $\mathcal{S} = \emptyset$
3: **for** $s \in \mathcal{D}$ **do**
4:     Set $\tilde{p}_s = \bar{q}\tilde{\sigma}^2(\mathbf{x}_s)$
5:     Draw $q_s \sim \text{Bernoulli}(\tilde{p}_s)$
6:     If $q_s = 1$, add $s$ into $\mathcal{S}$
7: **Output:** $\mathcal{S}$

---

Intuitively, in Algorithm 2, the clients first agree upon a common dictionary $\mathcal{S}_t$ that serves as a good representation of the whole dataset at the current time $t$, and then project their local data to the subspace spanned by this dictionary before communication, in order to avoid directly sending the raw data as in Algorithm 1. Then using the event-trigger, each client monitors the amount of new knowledge it has gained through interactions with the environment from last synchronization. When there is a sufficient amount of new knowledge, it will inform all the other clients to perform a synchronization. As we will show in the following section, the size of $\mathcal{S}_t$ scales linearly w.r.t. the maximum information gain $\gamma_{NT}$, and therefore it improves both the local computation efficiency on each client, and the communication efficiency during the global synchronization.

## 4.2 Theoretical Analysis

Denote the sequence of time steps when global synchronization is performed, i.e., the event $\mathcal{U}_t(D)$ in Eq (4) is true, as $\{t_p\}_{p=1}^B$, where $B \in [NT]$ denotes the total number of global synchronizations. Note that in Algorithm 2, the dictionary is only updated during global synchronization, e.g., at time $t_p$, the dictionary $\{(\mathbf{x}_s, y_s)\}_{s \in \mathcal{S}_{t_p}}$ is sampled from the whole dataset $\{(\mathbf{x}_s, y_s)\}_{s \in [t_p]}$ in a distributed manner, and remains fixed for all the interactions happened at $t \in [t_p + 1, t_{p+1}]$. Moreover, at time $t_p$, all the clients synchronize their embedded statistics, so that $\mathcal{D}_{t_p}(i) = [t_p], \forall i \in [N]$.

Since Algorithm 2 enables local update on each client, for time step $t \in [t_p + 1, t_p]$, new data points are collected and added into $\mathcal{D}_t(i)$, such that $\mathcal{D}_t(i) \supseteq [t_p]$. This *decreases* the approximation accuracy of $\mathcal{S}_{t_p}$, as new data points may not be well approximated by $\mathcal{S}_{t_p}$. For example, in extreme cases, the new data could be orthogonal to the dictionary. To formally analyze the accuracy of the dictionary, we adopt the definition of $\epsilon$-accuracy from [5]. Denote by $\bar{\mathbf{S}}_{t,i} \in \mathbb{R}^{|\mathcal{D}_t(i)| \times |\mathcal{D}_t(i)|}$ a diagonal matrix, with its $s$-th diagonal entry equal to $\frac{1}{\sqrt{\tilde{p}_s}}$ if $s \in \mathcal{S}_{t_p}$ and 0 otherwise. Then if

$$(1 - \epsilon_{t,i})(\mathbf{\Phi}_{\mathcal{D}_t(i)}^\top \mathbf{\Phi}_{\mathcal{D}_t(i)} + \lambda \mathbf{I}) \preceq \mathbf{\Phi}_{\mathcal{D}_t(i)}^\top \bar{\mathbf{S}}_{t,i}^\top \bar{\mathbf{S}}_{t,i} \mathbf{\Phi}_{\mathcal{D}_t(i)} + \lambda \mathbf{I} \preceq (1 + \epsilon_{t,i})(\mathbf{\Phi}_{\mathcal{D}_t(i)}^\top \mathbf{\Phi}_{\mathcal{D}_t(i)} + \lambda \mathbf{I}),$$

we say the dictionary $\{(\mathbf{x}_s, y_s)\}_{s \in \mathcal{S}_{t_p}}$ is $\epsilon_{t,i}$-accurate w.r.t. dataset $\{(\mathbf{x}_s, y_s)\}_{s \in \mathcal{D}_t(i)}$.

As shown below, the accuracy of the dictionary for Nyström approximation is essential as it affects the width of the confidence ellipsoid, and thus affects the cumulative regret. Intuitively, in order to ensure its accuracy throughout the learning process, we need to 1) make sure the RLS procedure in line 8 of Algorithm 2 that happens at each global synchronization produces a representative set of data samples, and 2) monitor the extent to which the dictionary obtained in previous global synchronization has degraded over time, and when necessary, trigger a new global synchronization to update it. Compared with prior work that freezes the model in-between consecutive communications [4], the analysis of $\epsilon$-accuracy for Approx-DisKernelUCB is unique to our paper and the result is presented below.

**Lemma 4.1.** *With $\bar{q} = 6\frac{1+\epsilon}{1-\epsilon} \log(4NT/\delta)/\epsilon^2$, for some $\epsilon \in [0, 1)$, and threshold $D > 0$, Algorithm 2 guarantees that the dictionary is accurate with constant $\epsilon_{t,i} := \left(\epsilon + 1 - \frac{1}{1 + \frac{1+\epsilon}{1-\epsilon}D}\right)$, and its size $|\mathcal{S}_t| = O(\gamma_{NT})$ for all $t \in [NT]$.*

Based on Lemma 4.1, we can construct the following confidence ellipsoid for unknown parameter $\theta_\star$.

**Lemma 4.2** (Confidence Ellipsoid of Approximated Estimator). *Under the condition that $\bar{q} = 6\frac{1+\epsilon}{1-\epsilon}\log(4NT/\delta)/\epsilon^2$, for some $\epsilon \in [0,1)$, and threshold $D > 0$, with probability at least $1 - \delta$, we have $\forall t, i$ that*

$$\|\tilde{\theta}_{t,i} - \theta_\star\|_{\tilde{\mathbf{A}}_{t,i}} \leq \Big( \frac{1}{\sqrt{-\epsilon + 1/(\frac{1+\epsilon}{1-\epsilon}D)}} + 1 \Big)\sqrt{\lambda}\|\theta_\star\| + 2R\sqrt{\ln NT/\delta + \gamma_{NT}} := \alpha_{t,i}.$$

Using Lemma 4.2, we obtain the regret and communication cost upper bound of Approx-DisKernelUCB, which is given in Theorem 4.3 below.

**Theorem 4.3** (Regret and Communication Cost of Approx-DisKernelUCB). *Under the same condition as Lemma 4.2, and by setting $D = \frac{1}{N}, \epsilon < \frac{1}{3}$, we have*

$$R_{NT} = O\big(\sqrt{NT}(\|\theta_\star\|\sqrt{\gamma_{NT}} + \gamma_{NT})\big)$$

*with probability at least $1 - \delta$, and*

$$C_{NT} = O\big(N^2\gamma_{NT}^3\big)$$

Here we provide a proof sketch for Theorem 4.3, and the complete proof can be found in appendix.

*Proof Sketch.* Similar to the analysis of DisKernelUCB in Section 3.2 and DisLinUCB from [30], the cumulative regret incurred by Approx-DisKernelUCB can be decomposed in terms of 'good' and 'bad' epochs, and bounded separately. Here an epoch refers to the time period in-between two consecutive global synchronizations, e.g., the $p$-th epoch refers to $[t_{p-1} + 1, t_p]$. Now consider an imaginary centralized agent that has immediate access to each data point in the learning system, and denote by $A_t = \sum_{s=1}^{t} \phi_s\phi_s^\top$ for $t \in [NT]$ the matrix constructed by this centralized agent. We call the $p$-th epoch a good epoch if $\ln\big(\frac{\det(\mathbf{I}+\lambda^{-1}\mathbf{K}_{[t_p],[t_p]})}{\det(\mathbf{I}+\lambda^{-1}\mathbf{K}_{[t_{p-1}],[t_{p-1}]})}\big) \leq 1$, otherwise it is a bad epoch. Note that $\ln\big(\frac{\det(\mathbf{I}+\lambda^{-1}\mathbf{K}_{[t_1],[t_1]})}{\det(\mathbf{I})}\big) + \ln\big(\frac{\det(\mathbf{I}+\lambda^{-1}\mathbf{K}_{[t_2],[t_2]})}{\det(\mathbf{I}+\lambda^{-1}\mathbf{K}_{[t_1],[t_1]})}\big) + \cdots + \ln\big(\frac{\det(\mathbf{I}+\lambda^{-1}\mathbf{K}_{[NT],[NT]})}{\det(\mathbf{I}+\lambda^{-1}\mathbf{K}_{[t_B],[t_B]})}\big) = \ln(\det(\mathbf{I}+\lambda^{-1}\mathbf{K}_{[NT],[NT]})) \leq 2\gamma_{NT}$, where the last equality is due to the matrix determinant lemma, and the last inequality is by the definition of the maximum information gain $\gamma_{NT}$ in Lemma 3.1. Then based on the pigeonhole principle, there can be at most $2\gamma_{NT}$ bad epochs.

By combining Lemma E.1 and Lemma 4.2, we can bound the cumulative regret incurred during all good epochs, i.e., $R_{good} = O(\sqrt{NT}\gamma_{NT})$, which matches the optimal regret attained by the KernelUCB algorithm in centralized setting. Our analysis deviates from that of DisKernelUCB in the bad epochs, because of the difference in the event-trigger. Previously, the event-trigger of DisKernelUCB directly bounds the cumulative regret each client incurs during a bad epoch, i.e., $\sum_{t \in \mathcal{D}_{t_p}(i)\backslash\mathcal{D}_{t_{p-1}}(i)} \hat{\sigma}_{t-1,i}(\mathbf{x}_t) \leq \sqrt{(|\mathcal{D}_{t_p}(i)| - |\mathcal{D}_{t_{p-1}}(i)|)\log\big(\det(\mathbf{I}+\lambda^{-1}\mathbf{K}_{\mathcal{D}_t(i_t),\mathcal{D}_t(i_t)})/\det(\mathbf{I}+\lambda^{-1}\mathbf{K}_{\mathcal{D}_t(i_t)\backslash\Delta\mathcal{D}_t(i_t),\mathcal{D}_t(i_t)\backslash\Delta\mathcal{D}_t(i_t)})\big)} < \sqrt{D}$. However, the event trigger of Approx-DisKernelUCB only bounds part of it, i.e., $\sum_{t \in \mathcal{D}_{t_p}(i)\backslash\mathcal{D}_{t_{p-1}}(i)} \tilde{\sigma}_{t-1,i}(\mathbf{x}_t) \leq \sqrt{(|\mathcal{D}_{t_p}(i)| - |\mathcal{D}_{t_{p-1}}(i)|)D}$, which leads to $R_{bad} = O(\sqrt{T}\gamma_{NT}N\sqrt{D})$ that is slightly worse than that of DisKernelUCB, i.e., a $\sqrt{T}$ factor in place of the $\sqrt{\gamma_{NT}}$ factor. By setting $D = 1/N$, we have $R_{NT} = O(\sqrt{NT}\gamma_{NT})$. Note that, to make sure $\epsilon_{t,i} = \big(\epsilon + 1 - \frac{1}{1+\frac{1+\epsilon}{1-\epsilon}\frac{1}{N}}\big) \in [0,1)$ is still well-defined, we can set $\epsilon < 1/3$.

For communication cost analysis, we bound the total number of epochs $B$ by upper bounding the total number of summations like $\sum_{s=t_{p-1}+1}^{t_p} \hat{\sigma}_{t_{p-1}}^2(\mathbf{x}_s)$, over the time horizon $NT$. Using Lemma E.1, our event-trigger in Eq (4) provides a lower bound $\sum_{s=t_{p-1}+1}^{t_p} \hat{\sigma}_{t_{p-1}}^2(\mathbf{x}_s) \geq \frac{1-\epsilon}{1+\epsilon}D$. Then in order to apply the pigeonhole principle, we continue to upper bound the summation over all epochs, $\sum_{p=1}^{B} \sum_{s=t_{p-1}+1}^{t_p} \hat{\sigma}_{t_{p-1}}^2(\mathbf{x}_s) = \sum_{p=1}^{B} \sum_{s=t_{p-1}+1}^{t_p} \hat{\sigma}_{s-1}^2(\mathbf{x}_s)\frac{\hat{\sigma}_{t_{p-1}}^2(\mathbf{x}_s)}{\hat{\sigma}_{s-1}^2(\mathbf{x}_s)}$ by deriving a uniform bound for the ratio $\frac{\hat{\sigma}_{t_{p-1}}^2(\mathbf{x}_s)}{\hat{\sigma}_{s-1}^2(\mathbf{x}_s)} \leq \frac{\hat{\sigma}_{t_{p-1}}^2(\mathbf{x}_s)}{\hat{\sigma}_{t_p}^2(\mathbf{x}_s)} \leq 1 + \sum_{s=t_{p-1}+1}^{t_p} \hat{\sigma}_{t_{p-1}}^2(\mathbf{x}_s) \leq 1 + \frac{1+\epsilon}{1-\epsilon}\sum_{s=t_{p-1}+1}^{t_p} \tilde{\sigma}_{t_{p-1}}^2(\mathbf{x}_s)$ in terms of the communication threshold $D$ on each client. This leads to the following upper bound about the total number of epochs $B \leq \frac{1+\epsilon}{1-\epsilon}[\frac{1}{D} + \frac{1+\epsilon}{1-\epsilon}(N + L^2/(\lambda D))]2\gamma_{NT}$, and with $D = 1/N$, we have $C_{NT} \leq B \cdot N\gamma_{NT}^2 = O(N^2\gamma_{NT}^3)$, which completes the proof. $\qquad\square$

**Remark 2.** *Compared with DisKernelUCB's $O(TN^2d)$ communication cost, Approx-DisKernelUCB removes the linear dependence on $T$, but introduces an additional $\gamma_{NT}^3$ dependence due to the communication of the embedded statistics. In situations where $\gamma_{NT} \ll T^{1/3}d^{1/3}$, DisKernelUCB is preferable. As mentioned in Lemma 3.1, the value of $\gamma_{NT}$, which affects how much the data can be compressed, depends on the specific arm set of the problem and the kernel function of the choice. By Mercer's Theorem, one can represent the kernel using its eigenvalues, and $\gamma_{NT}$ characterizes how fast its eigenvalues decay. Vakili et al. [28] showed that for kernels whose eigenvalues decay exponentially, i.e., $\lambda_m = O(\exp(-m^{\beta_e}))$, for some $\beta_e > 0$, $\gamma_{NT} = O(\log^{1+\frac{1}{\beta_e}}(NT))$. In this case, Approx-DisKernelUCB is far more efficient than DisKernelUCB. This includes Gaussian kernel, which is widely used for GPs and SVMs. For kernels that have polynomially decaying eigenvalues, i.e., $\lambda_m = O(m^{-\beta_p})$, for some $\beta_p > 1$, $\gamma_{NT} = O(T^{\frac{1}{\beta_p}}\log^{1-\frac{1}{\beta_p}}(NT))$. Then as long as $\beta_p > 3$, Approx-DisKernelUCB still enjoys reduced communication cost.*

## 5 Experiments

In order to evaluate Approx-DisKernelUCB's effectiveness in reducing communication cost, we performed extensive empirical evaluations on both synthetic and real-world datasets, and the results (averaged over 3 runs) are reported in Figure 1, 2 and 3, respectively. We included DisKernelUCB, DisLinUCB [30], OneKernelUCB, and NKernelUCB [6] as baselines, where One-KernelUCB learns a shared bandit model across all clients' aggregated data where data aggregation happens immediately after each new data point becomes available, and N-KernelUCB learns a separated bandit model for each client with no communication. For all the kernelized algorithms, we used the Gaussian kernel $k(x, y) = \exp(-\gamma\|x - y\|^2)$. We did a grid search of $\gamma \in \{0.1, 1, 4\}$ for kernelized algorithms, and set $D = 20$ for DisLinUCB and DisKernelUCB, $D = 5$ for Approx-DisKernelUCB. For all algorithms, instead of using their theoretically derived exploration coefficient $\alpha$, we followed the convention [20, 32] to use grid search for $\alpha$ in $\{0.1, 1, 4\}$. Due to space limit, here we only present the experiment results and discussions. Details about the experiment setup are presented in appendix.

When examining the experiment results presented in Figure 1, 2 and 3, we can first look at the cumulative regret and communication cost of OneKernelUCB and NKernelUCB, which correspond to the two extreme cases where the clients communicate in every time step to learn a shared model, and each client learns its own model independently with no communication, respectively. OneKernelUCB achieves the smallest cumulative regret in all experiments, while also incurring the highest communication cost, i.e., $O(TN^2d)$. This demonstrates the need of efficient data aggregation across clients for reducing regret. Second, we can observe that DisKernelUCB incurs the second highest communication cost in all experiments due to the transfer of raw data, as we have discussed in Remark 1, which makes it prohibitively expensive for distributed setting. On the other extreme, we can see that DisLinUCB incurs very small communication cost thanks to its closed-form solution, but fails to capture the complicated reward mappings in most of these datasets, e.g. in Figure 1(a), 2(b) and 3(a), it leads to even worse regret than NKernelUCB that learns a kernelized bandit model independently for each client. In comparison, the proposed Approx-DisKernelUCB algorithm enjoys the best of both worlds in most cases, i.e., it can take advantage of the superior modeling power of kernels to reduce regret, while only requiring a relatively low communication cost for clients to collaborate.

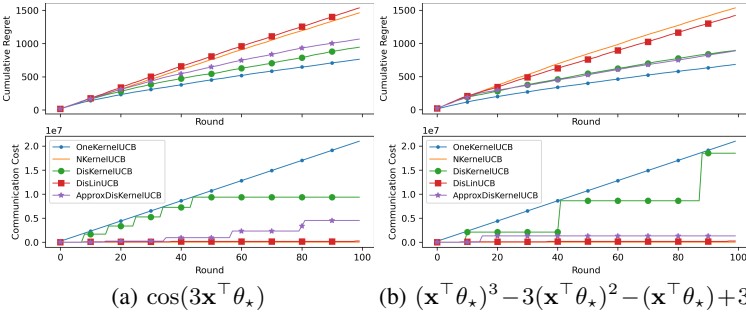

(a) $\cos(3\mathbf{x}^\top\theta_\star)$        (b) $(\mathbf{x}^\top\theta_\star)^3 - 3(\mathbf{x}^\top\theta_\star)^2 - (\mathbf{x}^\top\theta_\star) + 3$

Figure 1: Experiment results on synthetic datasets with different reward function $f(\mathbf{x})$.

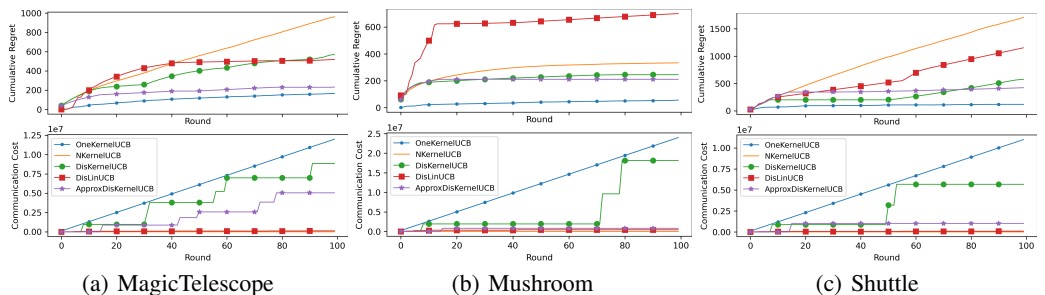

(a) MagicTelescope           (b) Mushroom           (c) Shuttle

Figure 2: Experiment results on UCI datasets.

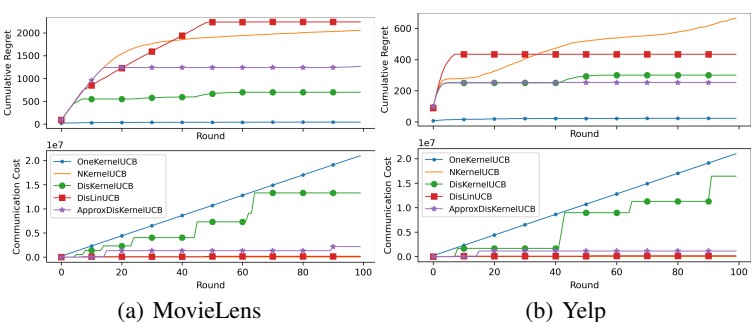

(a) MovieLens           (b) Yelp

Figure 3: Experiment results on MovieLens & Yelp datasets.

On all the datasets, Approx-DisKernelUCB achieved comparable regret with DisKernelUCB that maintains exact kernelized estimators, and sometimes even getting very close to OneKernelUCB, e.g., in Figure 1(b) and 2(a), but its communication cost is only slightly higher than that of DisLinUCB.

## Conclusion

In this paper, we proposed the first communication efficient algorithm for distributed kernel bandits using Nyström approximation. Clients in the learning system project their local data to a finite RKHS spanned by a shared dictionary, and then communicate the embedded statistics for collaborative exploration. To ensure communication efficiency, the frequency of dictionary update and synchronization of embedded statistics are controlled by an event-trigger. The algorithm is proved to incur $O(N^2\gamma_{NT}^3)$ communication cost, while attaining the optimal $O(\sqrt{NT}\gamma_{NT})$ cumulative regret.

We should note that the total number of synchronizations required by Approx-DisKernelUCB is $N\gamma_{NT}$, which is $\sqrt{N}$ worse than DisKernelUCB. An important future direction of this work is to investigate whether this part can be further improved. The lower bound analysis for the communication cost of distributed contextual bandits still remains an open problem, and is an important future direction. To the best of our knowledge, the only applicable lower bound states that, in order to have smaller regret than the trivial $O(N\sqrt{T})$ result, i.e., run $N$ instances of optimal bandit algorithm with no communication, $\Omega(N)$ communications is necessary [14]. In comparison, it is more interesting to know what the communication lower bound is in order to attain the optimal $O(\sqrt{NT})$ regret. It is also interesting to extend our algorithm to P2P setting, i.e., no central server to coordinate the update of the shared dictionary and the exchange of embedded statistics, which may require utilizing local structures in the network of clients, to approximate each block of the kernel matrix separately [25].

## Acknowledgement

This work is supported by NSF grants IIS-2213700, IIS-2128019, IIS-1838615, IIS-2107304, CMMI-1653435, DMS-1953686, and ONR grant 1006977.

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
