# A Technical Lemmas

**Lemma A.1** (Lemma 12 of [1]). *Let $A$, $B$ and $C$ be positive semi-definite matrices with finite dimension, such that $A = B + C$. Then, we have that:*

$$\sup_{\mathbf{x} \neq \mathbf{0}} \frac{\mathbf{x}^\top A \mathbf{x}}{\mathbf{x}^\top B \mathbf{x}} \leq \frac{\det(A)}{\det(B)}$$

**Lemma A.2** (Extension of Lemma A.1 to kernel matrix). *Define positive definite matrices $A = \lambda \mathbf{I} + \mathbf{\Phi}_1^\top \mathbf{\Phi}_1 + \mathbf{\Phi}_2^\top \mathbf{\Phi}_2$ and $B = \lambda \mathbf{I} + \mathbf{\Phi}_1^\top \mathbf{\Phi}_1$, where $\mathbf{\Phi}_1^\top \mathbf{\Phi}_1, \mathbf{\Phi}_2^\top \mathbf{\Phi}_2 \in \mathbb{R}^{p \times p}$ and $p$ is possibly infinite. Then, we have that:*

$$\sup_{\phi \neq \mathbf{0}} \frac{\phi^\top A \phi}{\phi^\top B \phi} \leq \frac{\det(\mathbf{I} + \lambda^{-1}\mathbf{K}_A)}{\det(\mathbf{I} + \lambda^{-1}\mathbf{K}_B)}$$

*where $\mathbf{K}_A = \begin{bmatrix} \mathbf{\Phi}_1 \\ \mathbf{\Phi}_2 \end{bmatrix} \begin{bmatrix} \mathbf{\Phi}_1^\top, \mathbf{\Phi}_2^\top \end{bmatrix}$ and $\mathbf{K}_B = \mathbf{\Phi}_1 \mathbf{\Phi}_1^\top$.*

*Proof of Lemma A.2.* Similar to the proof of Lemma 12 of [1], we start from the simple case when $\mathbf{\Phi}_2^\top \mathbf{\Phi}_2 = mm^\top$, where $m \in \mathbb{R}^p$. Using Cauchy-Schwartz inequality, we have

$$(\phi^\top m)^2 = (\phi^\top B^{1/2} B^{-1/2} m)^2 \leq \|B^{1/2}\phi\|^2 \|B^{-1/2}m\|^2 = \|\phi\|_B^2 \|m\|_{B^{-1}}^2,$$

and thus,

$$\phi^\top (B + mm^\top)\phi \leq \phi^\top B\phi + \|\phi\|_B^2 \|m\|_{B^{-1}}^2 = (1 + \|m\|_{B^{-1}}^2)\|\phi\|_B^2,$$

so we have

$$\frac{\phi^\top A \phi}{\phi^\top B \phi} \leq 1 + \|m\|_{B^{-1}}^2$$

for any $\phi$. Then using the kernel trick, e.g., see the derivation of Eq (27) in [31], we have

$$1 + \|m\|_{B^{-1}}^2 = \frac{\det(\mathbf{I} + \lambda^{-1}\mathbf{K}_A)}{\det(\mathbf{I} + \lambda^{-1}\mathbf{K}_B)},$$

which finishes the proof of this simple case. Now consider the general case where $\mathbf{\Phi}_2^\top \mathbf{\Phi}_2 = m_1 m_1^\top + m_2 m_2^\top + \cdots + m_{t-1} m_{t-1}^\top$. Let's define $V_s = B + m_1 m_1^\top + m_2 m_2^\top + \cdots + m_{s-1} m_{s-1}^\top$ and the corresponding kernel matrix $\mathbf{K}_{V_s} = \begin{bmatrix} \mathbf{\Phi}_1 \\ m_1^\top \\ \cdots \\ m_{s-1}^\top \end{bmatrix} \begin{bmatrix} \mathbf{\Phi}_1^\top, m_1, \ldots, m_{s-1} \end{bmatrix}$, and note that $\frac{\phi^\top A \phi}{\phi^\top B \phi} = \frac{\phi^\top V_t \phi}{\phi^\top V_{t-1}\phi} \frac{\phi^\top V_{t-1}\phi}{\phi^\top V_{t-2}\phi} \cdots \frac{\phi^\top V_2 \phi}{\phi^\top B \phi}$. Then we can apply the result for the simple case on each term in the product above, which gives us

$$\frac{\phi^\top A \phi}{\phi^\top B \phi} \leq \frac{\det(\mathbf{I} + \lambda^{-1}\mathbf{K}_{V_t})}{\det(\mathbf{I} + \lambda^{-1}\mathbf{K}_{V_{t-1}})} \frac{\det(\mathbf{I} + \lambda^{-1}\mathbf{K}_{V_{t-1}})}{\det(\mathbf{I} + \lambda^{-1}\mathbf{K}_{V_{t-2}})} \cdots \frac{\det(\mathbf{I} + \lambda^{-1}\mathbf{K}_{V_2})}{\det(\mathbf{I} + \lambda^{-1}\mathbf{K}_B)}$$

$$= \frac{\det(\mathbf{I} + \lambda^{-1}\mathbf{K}_{V_t})}{\det(\mathbf{I} + \lambda^{-1}\mathbf{K}_B)} = \frac{\det(\mathbf{I} + \lambda^{-1}\mathbf{K}_A)}{\det(\mathbf{I} + \lambda^{-1}\mathbf{K}_B)},$$

which finishes the proof.

$\square$

**Lemma A.3** (Eq (26) and Eq (27) of [31]). *Let $\{\phi_t\}_{t=1}^\infty$ be a sequence in $\mathbb{R}^p$, $V \in \mathbb{R}^{p \times p}$ a positive definite matrix, where $p$ is possibly infinite, and define $V_t = V + \sum_{s=1}^t \phi_s \phi_s^\top$. Then we have that*

$$\sum_{t=1}^n \min\left(\|\phi_t\|_{V_{t-1}^{-1}}^2, 1\right) \leq 2\ln\left(\det(\mathbf{I} + \lambda^{-1}\mathbf{K}_{V_t})\right),$$

*where $\mathbf{K}_{V_t}$ is the kernel matrix corresponding to $V_t$ as defined in Lemma A.2.*

**Lemma A.4** (Lemma 4 of [4]). *For $t > t_{last}$, we have for any $\mathbf{x} \in \mathbb{R}^d$*

$$\hat{\sigma}_t^2(\mathbf{x}) \leq \hat{\sigma}_{t_{last}}^2(\mathbf{x}) \leq \left(1 + \sum_{s=t_{last}+1}^t \hat{\sigma}_{t_{last}}^2(\mathbf{x}_s)\right)\hat{\sigma}_t^2(\mathbf{x})$$

# B  Confidence Ellipsoid for DisKernelUCB

In this section, we construct the confidence ellipsoid for DisKernelUCB as shown in Lemma B.1.

**Lemma B.1** (Confidence Ellipsoid for DisKernelUCB)**.** *Let $\delta \in (0, 1)$. With probability at least $1 - \delta$, for all $t \in [NT], i \in [N]$, we have*

$$\|\hat{\theta}_{t,i} - \theta_\star\|_{\mathbf{A}_{t,i}} \leq \sqrt{\lambda}\|\theta_\star\| + R\sqrt{2\ln(NT/\delta) + \ln(\det(\mathbf{K}_{\mathcal{D}_t(i),\mathcal{D}_t(i)}/\lambda + \mathbf{I}))}.$$

The analysis is rooted in [31] for kernelized contextual bandit, but with non-trivial extensions to address the dependencies due to the event-triggered distributed communication. This problem also exists in prior works of distributed linear bandit, but was not addressed rigorously (see Lemma H.1. of [30]). First, recall that the Ridge regression estimator

$$\hat{\theta}_{t,i} = \mathbf{A}_{t,i}^{-1} \sum_{s \in \mathcal{D}_t(i)} \phi_s y_s = \mathbf{A}_{t,i}^{-1} \sum_{s \in \mathcal{D}_t(i)} \phi_s (\phi_s^\top \theta_\star + \eta_s)$$

$$= \theta_\star - \lambda \mathbf{A}_{t,i}^{-1}\theta_\star + \mathbf{A}_{t,i}^{-1} \sum_{s \in \mathcal{D}_t(i)} \phi_s \eta_s,$$

and thus, we have

$$\begin{aligned}
\|\mathbf{A}_{t,i}^{1/2}(\hat{\theta}_{t,i} - \theta_\star)\| &= \|-\lambda\mathbf{A}_{t,i}^{-1/2}\theta_\star + \mathbf{A}_{t,i}^{-1/2} \sum_{s \in \mathcal{D}_t(i)} \phi_s \eta_s\| \\
&\leq \|\lambda\mathbf{A}_{t,i}^{-1/2}\theta_\star\| + \|\mathbf{A}_{t,i}^{-1/2} \sum_{s \in \mathcal{D}_t(i)} \phi_s \eta_s\| \\
&\leq \sqrt{\lambda}\|\theta_\star\| + \|\mathbf{A}_{t,i}^{-1/2} \sum_{s \in \mathcal{D}_t(i)} \phi_s \eta_s\|
\end{aligned} \tag{5}$$

where the first inequality is due to the triangle inequality, and the second is due to the property of Rayleigh quotient, i.e., $\|\mathbf{A}_{t,i}^{-1/2}\theta_\star\| \leq \|\theta_\star\|\sqrt{\lambda_{max}(\mathbf{A}_{t,i}^{-1})} \leq \|\theta_\star\|\frac{1}{\sqrt{\lambda}}$.

**Difference from standard argument**  Note that the second term may seem similar to the ones appear in the self-normalized bound in previous works of linear and kernelized bandits [1, 6, 31]. However, a main difference is that $\mathcal{D}_t(i)$, i.e., the sequence of indices for the data points used to update client $i$, is constructed using the event-trigger as defined in Eq (2) . The event-trigger is data-dependent, and thus it is a delayed and permuted version of the original sequence $[t]$. It is delayed in the sense that the length $|\mathcal{D}_t(i)| < t$ unless $t$ is the global synchronization step. It is permuted in the sense that every client receives the data in a different order, i.e., before the synchronization, each client first updates using its local new data, and then receives data from other clients at the synchronization. This prevents us from directly applying Lemma 3.1 of [31], and requires a more careful treatment as shown in the following paragraph.

First, we should note that during the time steps of global synchronization, i.e., $t \in \{t_p\}_{p \in [B]}$, we have $\mathcal{D}_t(i) = [t], \forall i \in [N]$, which recovers the case under centralized setting, i.e., the centralized agent that has access to all data points in the learning system. Therefore, analogous to the proof of RARELY SWITCHING OFUL algorithm in Appendix D of [1], with the standard argument in [31], we have

$$\|\mathbf{A}_{t,i}^{-1/2} \sum_{s \in \mathcal{D}_t(i)} \phi_s \eta_s\| \leq R\sqrt{2\ln(1/\delta) + \ln(\det(\mathbf{K}_{\mathcal{D}_t(i),\mathcal{D}_t(i)}/\lambda + \mathbf{I}))}$$

for all $t \in \{t_p\}_{p \in [B]}$ and $i \in [N]$, with probability at least $1 - \delta$. If our proposed algorithm has no local update, or use the 'hallucinating update' as in [4], then this would suffice. However, the existence of local update requires us to obtain self-normalized bounds for the local models that have been updated using each client's newly collected data after the synchronization step, which leads to the issue mentioned in the previous paragraph. Therefore, we need to address this issue by a union bound over all possible time steps of global synchronization and all clients.

Specifically, consider some time step $t \notin \{t_p\}_{p \in [B]}$ and client $i$. We denote the time step of the most recent global synchronization to $t$ as $t_{\text{last}}$, and define the filtration $\{\mathcal{F}_s\}_{s=0}^{t_{\text{last}}} \cup$

$\{\mathcal{F}_{s,i}\}_{s=t_{last}+1}^{\infty}$, where the $\sigma$-algebra $\mathcal{F}_s = \sigma\big((\mathbf{x}_\tau, \eta_\tau)_{\tau \in [s]}\big)$ for $s \in [0, t_{last}]$, and $\mathcal{F}_{s,i} = \sigma\big((\mathbf{x}_\tau, \eta_\tau)_{\tau \in [t_{last}]}, (\mathbf{x}_\tau, \eta_\tau)_{\tau \in [t_{last}, s], i_\tau = i}\big)$ for $s \geq t_{last} + 1$. By applying the standard argument for self-normalized bound using the filtration constructed above and then an union bound over $N$ clients, we have

$$\|\mathbf{A}_{t,i}^{-1/2} \sum_{s \in \mathcal{D}_t(i)} \phi_s \eta_s\| \leq R\sqrt{2\ln(N/\delta) + \ln(\det(\mathbf{K}_{\mathcal{D}_t(i), \mathcal{D}_t(i)}/\lambda + \mathbf{I}))}$$

for all $t > t_{last}$ and $i \in [N]$, with probability at least $1 - \delta$. As the time step of global synchronization $t_{last}$ is data-dependent, and thus can take any value in $[T]$, we apply another union bound, which finishes the proof.

## C    Proof of Lemma 3.1: Regret and Communication Cost of DisKernelUCB

Based on Lemma B.1 and the arm selection rule in Eq (1), we have

$$f(\mathbf{x}_t^\star) \leq \hat{\mu}_{t-1, i_t}(\mathbf{x}_t^\star) + \alpha_{t-1, i_t} \hat{\sigma}_{t-1, i_t}(\mathbf{x}_t^\star) \leq \hat{\mu}_{t-1, i_t}(\mathbf{x}_t) + \alpha_{t-1, i_t} \hat{\sigma}_{t-1, i_t}(\mathbf{x}_t),$$
$$f(\mathbf{x}_t) \geq \hat{\mu}_{t-1, i_t}(\mathbf{x}_t) - \alpha_{t-1, i_t} \hat{\sigma}_{t-1, i_t}(\mathbf{x}_t),$$

and thus $r_t = f(\mathbf{x}_t^\star) - f(\mathbf{x}_t) \leq 2\alpha_{t-1, i_t} \hat{\sigma}_{t-1, i_t}(\mathbf{x}_t)$, for all $t \in [NT]$, with probability at least $1 - \delta$. Then following similar steps as DisLinUCB of [30], we can obtain the regret and communication cost upper bound of DisKernelUCB.

### C.1    Regret Upper Bound

We call the time period in-between two consecutive global synchronizations as an epoch, i.e., the $p$-th epoch refers to $[t_{p-1} + 1, t_p]$, where $p \in [B]$ and $0 \leq B \leq NT$ denotes the total number of global synchronizations. Now consider an imaginary centralized agent that has immediate access to each data point in the learning system, and denote by $A_t = \sum_{s=1}^{t} \phi_s \phi_s^\top$ and $\mathbf{K}_{[t],[t]}$ for $t \in [NT]$ the covariance matrix and kernel matrix constructed by this centralized agent. Then similar to [30], we call the $p$-th epoch a good epoch if

$$\ln\left(\frac{\det(\mathbf{I} + \lambda^{-1}\mathbf{K}_{[t_p],[t_p]})}{\det(\mathbf{I} + \lambda^{-1}\mathbf{K}_{[t_{p-1}],[t_{p-1}]})}\right) \leq 1,$$

otherwise it is a bad epoch. Note that $\ln(\det(I + \lambda^{-1}\mathbf{K}_{[NT],[NT]})) \leq 2\gamma_{NT}$ by definition of $\gamma_{NT}$, i.e., the maximum information gain. Since $\ln(\frac{\det(\mathbf{I}+\lambda^{-1}\mathbf{K}_{[t_1],[t_1]})}{\det(\mathbf{I})}) + \ln(\frac{\det(\mathbf{I}+\lambda^{-1}\mathbf{K}_{[t_2],[t_2]})}{\det(\mathbf{I}+\lambda^{-1}\mathbf{K}_{[t_1],[t_1]})}) + \cdots + \ln(\frac{\det(\mathbf{I}+\lambda^{-1}\mathbf{K}_{[NT],[NT]})}{\det(\mathbf{I}+\lambda^{-1}\mathbf{K}_{[t_B],[t_B]})}) = \ln(\det(I + \lambda^{-1}\mathbf{K}_{[NT],[NT]})) \leq 2\gamma_{NT}$, and due to the pigeonhole principle, there can be at most $2\gamma_{NT}$ bad epochs.

If the instantaneous regret $r_t$ is incurred during a good epoch, we have

$$r_t \leq 2\alpha_{t-1, i_t} \|\phi_t\|_{\mathbf{A}_{t-1, i_t}^{-1}} \leq 2\alpha_{t-1, i_t} \|\phi_t\|_{\mathbf{A}_{t-1}^{-1}} \sqrt{\|\phi_t\|_{\mathbf{A}_{t-1, i_t}^{-1}}^2 / \|\phi_t\|_{\mathbf{A}_{t-1}^{-1}}^2}$$

$$= 2\alpha_{t-1, i_t} \|\phi_t\|_{\mathbf{A}_{t-1}^{-1}} \sqrt{\frac{\det(\mathbf{I} + \lambda^{-1}\mathbf{K}_{[t-1],[t-1]})}{\det(\mathbf{I} + \lambda^{-1}\mathbf{K}_{\mathcal{D}_{t-1}(i_t), \mathcal{D}_{t-1}(i_t)})}}$$

$$\leq 2\sqrt{e}\alpha_{t-1, i_t} \|\phi_t\|_{\mathbf{A}_{t-1}^{-1}}$$

where the second inequality is due to Lemma A.2, and the last inequality is due to the definition of good epoch, i.e., $\frac{\det(\mathbf{I}+\lambda^{-1}\mathbf{K}_{[t-1],[t-1]})}{\det(\mathbf{I}+\lambda^{-1}\mathbf{K}_{\mathcal{D}_{t-1}(i_t), \mathcal{D}_{t-1}(i_t)})} \leq \frac{\det(\mathbf{I}+\lambda^{-1}\mathbf{K}_{[t_p],[t_p]})}{\det(\mathbf{I}+\lambda^{-1}\mathbf{K}_{[t_{p-1}],[t_{p-1}]})} \leq e$. Define $\alpha_{NT} := \sqrt{\lambda}\|\theta_\star\| + \sqrt{2\ln(NT/\delta) + \ln(\det(\mathbf{K}_{[NT],[NT]}/\lambda + \mathbf{I}))}$. Then using standard arguments,

the cumulative regret incurred in all good epochs can be bounded by,

$$R_{good} = \sum_{p=1}^{B} \mathbb{1}\{\ln(\frac{\det(\mathbf{I} + \lambda^{-1}\mathbf{K}_{[t_p],[t_p]})}{\det(\mathbf{I} + \lambda^{-1}\mathbf{K}_{[t_{p-1}],[t_{p-1}]})}) \leq 1\} \sum_{t=t_{p-1}}^{t_p} r_t \leq \sum_{t=1}^{NT} 2\sqrt{e}\alpha_{t-1,i_t}\|\phi_t\|_{\mathbf{A}_{t-1}^{-1}}$$

$$\leq 2\sqrt{e}\alpha_{NT} \sum_{t=1}^{NT} \|\phi_t\|_{\mathbf{A}_{t-1}^{-1}} \leq 2\sqrt{e}\alpha_{NT}\sqrt{NT \cdot 2\ln(\det(\mathbf{I} + \lambda^{-1}\mathbf{K}_{[NT],[NT]}))}$$

$$\leq 2\sqrt{e}\alpha_{NT}\sqrt{NT \cdot 4\gamma_{NT}} = O\Big(\sqrt{NT}(\|\theta_\star\|\sqrt{\gamma_{NT}} + \gamma_{NT})\Big)$$

where the third inequality is due to Cauchy-Schwartz and Lemma A.3, and the forth is due to the definition of maximum information gain $\gamma_{NT}$.

Then we look at the regret incurred during bad epochs. Consider some bad epoch $p$, and the cumulative regret incurred during this epoch can be bounded by

$$\sum_{t=t_{p-1}+1}^{t_p} r_t = \sum_{i=1}^{N} \sum_{t\in\mathcal{D}_{t_p}(i)\setminus\mathcal{D}_{t_{p-1}}(i)} r_t \leq 2\alpha_{NT} \sum_{i=1}^{N} \sum_{t\in\mathcal{D}_{t_p}(i)\setminus\mathcal{D}_{t_{p-1}}(i)} \|\phi_t\|_{\mathbf{A}_{t-1,i}^{-1}}$$

$$\leq 2\alpha_{NT} \sum_{i=1}^{N} \sqrt{(|\mathcal{D}_{t_p}(i)| - |\mathcal{D}_{t_{p-1}}(i)|) \sum_{t\in\mathcal{D}_{t_p}(i)\setminus\mathcal{D}_{t_{p-1}}(i)} \|\phi_t\|_{\mathbf{A}_{t-1,i}^{-1}}^2}$$

$$\leq 2\alpha_{NT} \sum_{i=1}^{N} \sqrt{2(|\mathcal{D}_{t_p}(i)| - |\mathcal{D}_{t_{p-1}}(i)|) \ln(\frac{\det(\mathbf{I} + \lambda^{-1}\mathbf{K}_{\mathcal{D}_{t_p}(i),\mathcal{D}_{t_p}(i)})}{\det(\mathbf{I} + \lambda^{-1}K_{\mathcal{D}_{t_{p-1}}(i),\mathcal{D}_{t_{p-1}}(i)})})}$$

$$\leq 2\sqrt{2}\alpha_{NT}N\sqrt{D}$$

where the last inequality is due to our event-trigger in Eq (2). Since there can be at most $2\gamma_{NT}$ bad epochs, the cumulative regret incurred in all bad epochs

$$R_{bad} \leq 2\gamma_{NT} \cdot 2\sqrt{2}\alpha_{NT}N\sqrt{D} = O\Big(ND^{0.5}(\|\theta_\star\|\gamma_{NT} + \gamma_{NT}^{1.5})\Big)$$

Combining cumulative regret incurred during both good and bad epochs, we have

$$R_{NT} = R_{good} + R_{bad} = O\big((\sqrt{NT} + N\sqrt{D\gamma_{NT}})(\|\theta_\star\|\sqrt{\gamma_{NT}} + \gamma_{NT})\big)$$

## C.2 Communication Upper Bound

For some $\alpha > 0$, there can be at most $\lceil\frac{NT}{\alpha}\rceil$ epochs with length larger than $\alpha$. Based on our event-trigger design, we know that $(|\mathcal{D}_{t_p}(i_{t_p})| - |\mathcal{D}_{t_{p-1}}(i_{t_p})|) \ln(\frac{\det(\mathbf{I}+\lambda^{-1}\mathbf{K}_{[t_p],[t_p]})}{\det(\mathbf{I}+\lambda^{-1}K_{[t_{p-1}],[t_{p-1}]})}) \geq (|\mathcal{D}_{t_p}(i_{t_p})| - |\mathcal{D}_{t_{p-1}}(i_{t_p})|) \ln(\frac{\det(\mathbf{I}+\lambda^{-1}\mathbf{K}_{\mathcal{D}_{t_p}(i_{t_p}),\mathcal{D}_{t_p}(i_{t_p})})}{\det(\mathbf{I}+\lambda^{-1}K_{\mathcal{D}_{t_{p-1}}(i_{t_p}),\mathcal{D}_{t_{p-1}}(i_{t_p})})}) \geq D$ for any epoch $p \in [B]$, where $i_{t_p}$ is the client who triggers the global synchronization at time step $t_p$. Then if the length of certain epoch $p$ is smaller than $\alpha$, i.e., $t_p - t_{p-1} \leq \alpha$, we have $\ln(\frac{\det(\mathbf{I}+\lambda^{-1}\mathbf{K}_{[t_p],[t_p]})}{\det(\mathbf{I}+\lambda^{-1}\mathbf{K}_{[t_{p-1}],[t_{p-1}]})}) \geq \frac{ND}{\alpha}$. Since $\ln(\frac{\det(\mathbf{I}+\lambda^{-1}\mathbf{K}_{[t_1],[t_1]})}{\det(\mathbf{I})}) + \ln(\frac{\det(\mathbf{I}+\lambda^{-1}\mathbf{K}_{[t_2],[t_2]})}{\det(\mathbf{I}+\lambda^{-1}\mathbf{K}_{[t_1],[t_1]})}) + \cdots + \ln(\frac{\det(\mathbf{I}+\lambda^{-1}\mathbf{K}_{[t_B],[t_B]})}{\det(\mathbf{I}+\lambda^{-1}\mathbf{K}_{[t_{B-1}],[t_{B-1}]})}) \leq \ln(\det(\mathbf{I} + \lambda^{-1}\mathbf{K}_{[NT],[NT]})) \leq 2\gamma_{NT}$, the total number of such epochs is upper bounded by $\lceil\frac{2\gamma_{NT}\alpha}{ND}\rceil$. Combining the two terms, the total number of epochs can be bounded by,

$$B \leq \lceil\frac{NT}{\alpha}\rceil + \lceil\frac{2\gamma_{NT}\alpha}{ND}\rceil$$

where the LHS can be minimized using the AM-GM inequality, i.e., $B \leq \sqrt{\frac{NT}{\alpha}\frac{2\gamma_{NT}\alpha}{ND}} = \sqrt{\frac{2\gamma_{NT}T}{D}}$. To obtain the optimal order of regret, we set $D = O(\frac{T}{N\gamma_{NT}})$, so that $R_{NT} = O\big(\sqrt{NT}(\|\theta_\star\|\sqrt{\gamma_{NT}} + \gamma_{NT})\big)$. And the total number of epochs $B = O(\sqrt{N}\gamma_{NT})$. However, we should note that as DisKernelUCB communicates all the unshared raw data at each global synchronization, the total

communication cost mainly depends on when the last global synchronization happens. Since the sequence of candidate sets $\{\mathcal{A}_t\}_{t\in[NT]}$, which controls the growth of determinant, is an arbitrary subset of $\mathcal{A}$, the time of last global synchronization could happen at the last time step $t = NT$. Therefore, $C_T = O(N^2Td)$ in such a worst case.

# D   Derivation of the Approximated Mean and Variance in Section 4

For simplicity, subscript $t$ is omitted in this section. The approximated Ridge regression estimator for dataset $\{(\mathbf{x}_s, y_s)\}_{s\in\mathcal{D}}$ is formulated as

$$\tilde{\theta} = \arg\min_{\theta\in\mathcal{H}} \sum_{s\in\mathcal{D}} \left((\mathbf{P}_{\mathcal{S}}\phi_s)^\top\theta - y_s\right)^2 + \lambda\|\theta\|_2^2$$

where $\mathcal{D}$ denotes the sequence of time indices for data in the original dataset, $\mathcal{S}\subseteq\mathcal{D}$ denotes the time indices for data in the dictionary, and $\mathbf{P}_{\mathcal{S}}\in\mathbb{R}^{p\times p}$ denotes the orthogonal projection matrix defined by $\mathcal{S}$. Then by taking derivative and setting it to zero, we have $(\mathbf{P}_{\mathcal{S}}\mathbf{\Phi}_{\mathcal{D}}^\top\mathbf{\Phi}_{\mathcal{D}}\mathbf{P}_{\mathcal{S}} + \lambda\mathbf{I})\tilde{\theta} = \mathbf{P}_{\mathcal{S}}\mathbf{\Phi}_{\mathcal{D}}^\top\mathbf{y}_{\mathcal{D}}$, and thus $\tilde{\theta} = \tilde{\mathbf{A}}^{-1}\tilde{\mathbf{b}}$, where $\tilde{\mathbf{A}} = \mathbf{P}_{\mathcal{S}}\mathbf{\Phi}_{\mathcal{D}}^\top\mathbf{\Phi}_{\mathcal{D}}\mathbf{P}_{\mathcal{S}} + \lambda\mathbf{I}$ and $\tilde{\mathbf{b}} = \mathbf{P}_{\mathcal{S}}\mathbf{\Phi}_{\mathcal{D}}^\top\mathbf{y}_{\mathcal{D}}$.

Hence, the approximated mean reward and variance for some arm $\mathbf{x}$ are

$$\tilde{\mu}_{t,i}(\mathbf{x}) = \phi(\mathbf{x})^\top\tilde{\mathbf{A}}^{-1}\tilde{\mathbf{b}}$$

$$\tilde{\sigma}_{t,i}(\mathbf{x}) = \sqrt{\phi(\mathbf{x})^\top\tilde{\mathbf{A}}^{-1}\phi(\mathbf{x})}$$

To obtain their kernelized representation, we rewrite

$$(\mathbf{P}_{\mathcal{S}}\mathbf{\Phi}_{\mathcal{D}}^\top\mathbf{\Phi}_{\mathcal{D}}\mathbf{P}_{\mathcal{S}} + \lambda\mathbf{I})\tilde{\theta} = \mathbf{P}_{\mathcal{S}}\mathbf{\Phi}_{\mathcal{D}}^\top\mathbf{y}_{\mathcal{D}}$$
$$\Leftrightarrow \mathbf{P}_{\mathcal{S}}\mathbf{\Phi}_{\mathcal{D}}^\top(\mathbf{y}_{\mathcal{D}} - \mathbf{\Phi}_{\mathcal{D}}\mathbf{P}_{\mathcal{S}}\tilde{\theta}) = \lambda\tilde{\theta}$$
$$\Leftrightarrow \tilde{\theta} = \mathbf{P}_{\mathcal{S}}\mathbf{\Phi}_{\mathcal{D}}^\top\rho$$

where $\rho := \frac{1}{\lambda}(\mathbf{y}_{\mathcal{D}} - \mathbf{\Phi}_{\mathcal{D}}\mathbf{P}_{\mathcal{S}}\tilde{\theta}) = \frac{1}{\lambda}(\mathbf{y}_{\mathcal{D}} - \mathbf{\Phi}_{\mathcal{D}}\mathbf{P}_{\mathcal{S}}\mathbf{P}_{\mathcal{S}}\mathbf{\Phi}_{\mathcal{D}}^\top\rho)$. Solving this equation, we get $\rho = (\mathbf{\Phi}_{\mathcal{D}}\mathbf{P}_{\mathcal{S}}\mathbf{P}_{\mathcal{S}}\mathbf{\Phi}_{\mathcal{D}}^\top + \lambda\mathbf{I})^{-1}\mathbf{y}_{\mathcal{D}}$. Note that $\mathbf{P}_{\mathcal{S}}\mathbf{P}_{\mathcal{S}} = \mathbf{P}_{\mathcal{S}}$, since projection matrix $\mathbf{P}_{\mathcal{S}}$ is idempotent. Moreover, we have $(\mathbf{\Phi}^\top\mathbf{\Phi} + \lambda\mathbf{I})\mathbf{\Phi}^\top = \mathbf{\Phi}^\top(\mathbf{\Phi}\mathbf{\Phi}^\top + \lambda\mathbf{I})$, and $(\mathbf{\Phi}^\top\mathbf{\Phi} + \lambda\mathbf{I})^{-1}\mathbf{\Phi}^\top = \mathbf{\Phi}^\top(\mathbf{\Phi}\mathbf{\Phi}^\top + \lambda\mathbf{I})^{-1}$. Therefore, we can rewrite the approximated mean for some arm $\mathbf{x}$ as

$$\begin{aligned}
\tilde{\mu}(\mathbf{x}) &= \phi(\mathbf{x})^\top\mathbf{P}_{\mathcal{S}}\mathbf{\Phi}_{\mathcal{D}}^\top(\mathbf{\Phi}_{\mathcal{D}}\mathbf{P}_{\mathcal{S}}\mathbf{P}_{\mathcal{S}}\mathbf{\Phi}_{\mathcal{D}}^\top + \lambda\mathbf{I})^{-1}\mathbf{y}_{\mathcal{D}} \\
&= (\mathbf{P}_{\mathcal{S}}^{1/2}\phi(\mathbf{x}))^\top(\mathbf{\Phi}_{\mathcal{D}}\mathbf{P}_{\mathcal{S}}^{1/2})^\top[\mathbf{\Phi}_{\mathcal{D}}\mathbf{P}_{\mathcal{S}}^{1/2}(\mathbf{\Phi}_{\mathcal{D}}\mathbf{P}_{\mathcal{S}}^{1/2})^\top + \lambda\mathbf{I}]^{-1}\mathbf{y}_{\mathcal{D}} \\
&= (\mathbf{P}_{\mathcal{S}}^{1/2}\phi(\mathbf{x}))^\top(\mathbf{P}_{\mathcal{S}}^{1/2}\mathbf{\Phi}_{\mathcal{D}}^\top\mathbf{\Phi}_{\mathcal{D}}\mathbf{P}_{\mathcal{S}}^{1/2} + \lambda\mathbf{I})^{-1}(\mathbf{\Phi}_{\mathcal{D}}\mathbf{P}_{\mathcal{S}}^{1/2})^\top\mathbf{y}_{\mathcal{D}} \\
&= z(\mathbf{x};\mathcal{S})^\top(\mathbf{Z}_{\mathcal{D};\mathcal{S}}^\top\mathbf{Z}_{\mathcal{D};\mathcal{S}} + \lambda\mathbf{I})^{-1}\mathbf{Z}_{\mathcal{D};\mathcal{S}}^\top\mathbf{y}_{\mathcal{D}}
\end{aligned}$$

To derive the approximated variance, we start from the fact that $(\mathbf{P}_{\mathcal{S}}\mathbf{\Phi}_{\mathcal{D}}^\top\mathbf{\Phi}_{\mathcal{D}}\mathbf{P}_{\mathcal{S}} + \lambda\mathbf{I})\phi(\mathbf{x}) = \mathbf{P}_{\mathcal{S}}\mathbf{\Phi}_{\mathcal{D}}^\top\mathbf{\Phi}_{\mathcal{D}}\mathbf{P}_{\mathcal{S}}\phi(\mathbf{x}) + \lambda\phi(\mathbf{x})$, so

$$\begin{aligned}
\phi(\mathbf{x}) &= (\mathbf{P}_{\mathcal{S}}\mathbf{\Phi}_{\mathcal{D}}^\top\mathbf{\Phi}_{\mathcal{D}}\mathbf{P}_{\mathcal{S}} + \lambda\mathbf{I})^{-1}\mathbf{P}_{\mathcal{S}}\mathbf{\Phi}_{\mathcal{D}}^\top\mathbf{\Phi}_{\mathcal{D}}\mathbf{P}_{\mathcal{S}}\phi(\mathbf{x}) + \lambda(\mathbf{P}_{\mathcal{S}}\mathbf{\Phi}_{\mathcal{D}}^\top\mathbf{\Phi}_{\mathcal{D}}\mathbf{P}_{\mathcal{S}} + \lambda\mathbf{I})^{-1}\phi(\mathbf{x}) \\
&= \mathbf{P}_{\mathcal{S}}\mathbf{\Phi}_{\mathcal{D}}^\top(\mathbf{\Phi}_{\mathcal{D}}\mathbf{P}_{\mathcal{S}}\mathbf{P}_{\mathcal{S}}\mathbf{\Phi}_{\mathcal{D}}^\top + \lambda\mathbf{I})^{-1}\mathbf{\Phi}_{\mathcal{D}}\mathbf{P}_{\mathcal{S}}\phi(\mathbf{x}) + \lambda(\mathbf{P}_{\mathcal{S}}\mathbf{\Phi}_{\mathcal{D}}^\top\mathbf{\Phi}_{\mathcal{D}}\mathbf{P}_{\mathcal{S}} + \lambda\mathbf{I})^{-1}\phi(\mathbf{x})
\end{aligned}$$

Then we have

$$\begin{aligned}
&\phi(\mathbf{x})^\top\phi(\mathbf{x}) \\
=&\left\{\mathbf{P}_{\mathcal{S}}\mathbf{\Phi}_{\mathcal{D}}^\top(\mathbf{\Phi}_{\mathcal{D}}\mathbf{P}_{\mathcal{S}}\mathbf{P}_{\mathcal{S}}\mathbf{\Phi}_{\mathcal{D}}^\top + \lambda\mathbf{I})^{-1}\mathbf{\Phi}_{\mathcal{D}}\mathbf{P}_{\mathcal{S}}\phi(\mathbf{x}) + \lambda(\mathbf{P}_{\mathcal{S}}\mathbf{\Phi}_{\mathcal{D}}^\top\mathbf{\Phi}_{\mathcal{D}}\mathbf{P}_{\mathcal{S}} + \lambda\mathbf{I})^{-1}\phi(\mathbf{x})\right\}^\top \\
&\quad\left\{\mathbf{P}_{\mathcal{S}}\mathbf{\Phi}_{\mathcal{D}}^\top(\mathbf{\Phi}_{\mathcal{D}}\mathbf{P}_{\mathcal{S}}\mathbf{P}_{\mathcal{S}}\mathbf{\Phi}_{\mathcal{D}}^\top + \lambda\mathbf{I})^{-1}\mathbf{\Phi}_{\mathcal{D}}\mathbf{P}_{\mathcal{S}}\phi(\mathbf{x}) + \lambda(\mathbf{P}_{\mathcal{S}}\mathbf{\Phi}_{\mathcal{D}}^\top\mathbf{\Phi}_{\mathcal{D}}\mathbf{P}_{\mathcal{S}} + \lambda\mathbf{I})^{-1}\phi(\mathbf{x})\right\} \\
=&\phi(\mathbf{x})^\top\mathbf{P}_{\mathcal{S}}\mathbf{\Phi}_{\mathcal{D}}^\top(\mathbf{\Phi}_{\mathcal{D}}\mathbf{P}_{\mathcal{S}}\mathbf{P}_{\mathcal{S}}\mathbf{\Phi}_{\mathcal{D}}^\top + \lambda\mathbf{I})^{-1}\mathbf{\Phi}_{\mathcal{D}}\mathbf{P}_{\mathcal{S}}\mathbf{P}_{\mathcal{S}}\mathbf{\Phi}_{\mathcal{D}}^\top(\mathbf{\Phi}_{\mathcal{D}}\mathbf{P}_{\mathcal{S}}\mathbf{P}_{\mathcal{S}}\mathbf{\Phi}_{\mathcal{D}}^\top + \lambda\mathbf{I})^{-1}\mathbf{\Phi}_{\mathcal{D}}\mathbf{P}_{\mathcal{S}}\phi(\mathbf{x}) \\
&+ 2\lambda\phi(\mathbf{x})^\top\mathbf{P}_{\mathcal{S}}\mathbf{\Phi}_{\mathcal{D}}^\top(\mathbf{\Phi}_{\mathcal{D}}\mathbf{P}_{\mathcal{S}}\mathbf{P}_{\mathcal{S}}\mathbf{\Phi}_{\mathcal{D}}^\top + \lambda\mathbf{I})^{-1}\mathbf{\Phi}_{\mathcal{D}}\mathbf{P}_{\mathcal{S}}(\mathbf{P}_{\mathcal{S}}\mathbf{\Phi}_{\mathcal{D}}^\top\mathbf{\Phi}_{\mathcal{D}}\mathbf{P}_{\mathcal{S}} + \lambda\mathbf{I})^{-1}\phi(\mathbf{x}) \\
&+ \lambda\phi(\mathbf{x})^\top(\mathbf{P}_{\mathcal{S}}\mathbf{\Phi}_{\mathcal{D}}^\top\mathbf{\Phi}_{\mathcal{D}}\mathbf{P}_{\mathcal{S}} + \lambda\mathbf{I})^{-1}\lambda\mathbf{I}(\mathbf{P}_{\mathcal{S}}\mathbf{\Phi}_{\mathcal{D}}^\top\mathbf{\Phi}_{\mathcal{D}}\mathbf{P}_{\mathcal{S}} + \lambda\mathbf{I})^{-1}\phi(\mathbf{x}) \\
=&\phi(\mathbf{x})^\top\mathbf{P}_{\mathcal{S}}\mathbf{\Phi}_{\mathcal{D}}^\top(\mathbf{\Phi}_{\mathcal{D}}\mathbf{P}_{\mathcal{S}}\mathbf{P}_{\mathcal{S}}\mathbf{\Phi}_{\mathcal{D}}^\top + \lambda\mathbf{I})^{-1}\mathbf{\Phi}_{\mathcal{D}}\mathbf{P}_{\mathcal{S}}\phi(\mathbf{x}) + \lambda\phi(\mathbf{x})^\top(\mathbf{P}_{\mathcal{S}}\mathbf{\Phi}_{\mathcal{D}}^\top\mathbf{\Phi}_{\mathcal{D}}\mathbf{P}_{\mathcal{S}} + \lambda\mathbf{I})^{-1}\phi(\mathbf{x})
\end{aligned}$$

By rearranging terms, we have

$$
\begin{aligned}
\tilde{\sigma}^2(\mathbf{x}) =& \phi(\mathbf{x})^\top (\mathbf{P}_{\mathcal{S}} \mathbf{\Phi}_{\mathcal{D}}^\top \mathbf{\Phi}_{\mathcal{D}} \mathbf{P}_{\mathcal{S}} + \lambda \mathbf{I})^{-1} \phi(\mathbf{x}) \\
=& \frac{1}{\lambda} \{ \phi(\mathbf{x})^\top \phi(\mathbf{x}) - \phi(\mathbf{x})^\top \mathbf{P}_{\mathcal{S}} \mathbf{\Phi}_{\mathcal{D}}^\top (\mathbf{\Phi}_{\mathcal{D}} \mathbf{P}_{\mathcal{S}} \mathbf{P}_{\mathcal{S}} \mathbf{\Phi}_{\mathcal{D}}^\top + \lambda \mathbf{I})^{-1} \mathbf{\Phi}_{\mathcal{D}} \mathbf{P}_{\mathcal{S}} \phi(\mathbf{x}) \} \\
=& \frac{1}{\lambda} \{ k(\mathbf{x}, \mathbf{x}) - z(\mathbf{x}; \mathcal{S})^\top \mathbf{Z}_{\mathcal{D};\mathcal{S}}^\top \mathbf{Z}_{\mathcal{D};\mathcal{S}} [\mathbf{Z}_{\mathcal{D};\mathcal{S}}^\top \mathbf{Z}_{\mathcal{D};\mathcal{S}} + \lambda \mathbf{I}]^{-1} z(\mathbf{x}|\mathcal{S}) \}
\end{aligned}
$$

# E  Proof of Lemma 4.1

In the following, we analyze the $\epsilon_{t,i}$-accuracy of the dictionary for all $t, i$.

At the time steps when global synchronization happens, i.e., $t_p$ for $p \in [B]$, $\mathcal{S}_{t_p}$ is sampled from $[t_p] = \mathcal{D}_{t_p}(i)$ using approximated variance $\tilde{\sigma}_{t_{p-1},i}^2$. In this case, the accuracy of the dictionary only depends on the RLS procedure, and Calandriello et al. [4] have already showed that the following guarantee on the accuracy and size of dictionary holds $\forall t \in \{t_p\}_{p \in [B]}$.

**Lemma E.1** (Lemma 2 of [4]). *Under the condition that $\bar{q} = 6\frac{1+\epsilon}{1-\epsilon} \log(4NT/\delta)/\epsilon^2$, for some $\epsilon \in [0, 1)$, with probability at least $1 - \delta$, we have $\forall t \in \{t_p\}_{p \in [B]}$ that the dictionary $\{(\mathbf{x}_s, y_s)\}_{s \in \mathcal{S}_t}$ is $\epsilon$-accurate w.r.t. $\{(\mathbf{x}_s, y_s)\}_{s \in \mathcal{D}_t(i)}$, and $\frac{1-\epsilon}{1+\epsilon} \sigma_t^2(\mathbf{x}) \leq \tilde{\sigma}_t^2(\mathbf{x}) \leq \frac{1+\epsilon}{1-\epsilon} \sigma_t^2(\mathbf{x}), \forall \mathbf{x} \in \mathcal{A}$. Moreover, the size of dictionary $|\mathcal{S}_t| \leq 3(1 + L^2/\lambda) \frac{1+\epsilon}{1-\epsilon} \bar{q} \tilde{d}$, where $\tilde{d} := Tr(\mathbf{K}_{[NT],[NT]} (\mathbf{K}_{[NT],[NT]} + \lambda \mathbf{I})^{-1})$ denotes the effective dimension of the problem, and it is known that $\tilde{d} = O(\gamma_{NT})$ [6].*

Lemma E.1 guarantees that for all $t \in \{t_p\}_{p \in [B]}$, the dictionary has a constant accuracy, i.e., $\epsilon_{t,i} = \epsilon, \forall i$. In addition, since the dictionary is fixed for $t \notin \{t_p\}_{p \in [B]}$, its size $\mathcal{S}_t = O(\gamma_{NT}), \forall t \in [NT]$.

Then for time steps $t \notin \{t_p\}_{p \in [B]}$, due to the local update, the accuracy of the dictionary will degrade. However, thanks to our event-trigger in Eq (4), the extent of such degradation can be controlled, i.e., a new dictionary update will be triggered before the previous dictionary becomes completely irrelevant. This is shown in Lemma E.2 below.

**Lemma E.2.** *Under the condition that $\{(\mathbf{x}_s, y_s)\}_{s \in \mathcal{S}_{t_p}}$ is $\epsilon$-accurate w.r.t. $\{(\mathbf{x}_s, y_s)\}_{s \in \mathcal{D}_{t_p}(i)}$, $\forall t \in [t_p + 1, t_{p+1}], i \in [N]$, $\mathcal{S}_{t_p}$ is $\left(\epsilon + 1 - \frac{1}{1 + \frac{1+\epsilon}{1-\epsilon} D}\right)$-accurate w.r.t. $\mathcal{D}_t(i)$.*

Combining Lemma E.1 and Lemma E.2 finishes the proof.

*Proof of Lemma E.2.* Similar to [3], we can rewrite the $\epsilon$-accuracy condition of $\mathcal{S}_{t_p}$ w.r.t. $\mathcal{D}_t(i)$ for $t \in [t_p + 1, t_{p+1}]$ as

$$
\begin{aligned}
& (1 - \epsilon_{t,i})(\mathbf{\Phi}_{\mathcal{D}_t(i)}^\top \mathbf{\Phi}_{\mathcal{D}_t(i)} + \lambda \mathbf{I}) \preceq \mathbf{\Phi}_{\mathcal{D}_t(i)}^\top \bar{\mathbf{S}}_{t,i}^\top \bar{\mathbf{S}}_{t,i} \mathbf{\Phi}_{\mathcal{D}_t(i)} + \lambda \mathbf{I} \preceq (1 + \epsilon_{t,i})(\mathbf{\Phi}_{\mathcal{D}_t(i)}^\top \mathbf{\Phi}_{\mathcal{D}_t(i)} + \lambda \mathbf{I}) \\
\Leftrightarrow& -\epsilon_{t,i}(\mathbf{\Phi}_{\mathcal{D}_t(i)}^\top \mathbf{\Phi}_{\mathcal{D}_t(i)} + \lambda \mathbf{I}) \preceq \mathbf{\Phi}_{\mathcal{D}_t(i)}^\top \bar{\mathbf{S}}_{t,i}^\top \bar{\mathbf{S}}_{t,i} \mathbf{\Phi}_{\mathcal{D}_t(i)} - \mathbf{\Phi}_{\mathcal{D}_t(i)}^\top \mathbf{\Phi}_{\mathcal{D}_t(i)} \preceq \epsilon_{t,i}(\mathbf{\Phi}_{\mathcal{D}_t(i)}^\top \mathbf{\Phi}_{\mathcal{D}_t(i)} + \lambda \mathbf{I}) \\
\Leftrightarrow& -\epsilon_{t,i} \mathbf{I} \preceq (\mathbf{\Phi}_{\mathcal{D}_t(i)}^\top \mathbf{\Phi}_{\mathcal{D}_t(i)} + \lambda \mathbf{I})^{-1/2} (\mathbf{\Phi}_{\mathcal{D}_t(i)}^\top \bar{\mathbf{S}}_{t,i}^\top \bar{\mathbf{S}}_{t,i} \mathbf{\Phi}_{\mathcal{D}_t(i)} - \mathbf{\Phi}_{\mathcal{D}_t(i)}^\top \mathbf{\Phi}_{\mathcal{D}_t(i)})(\mathbf{\Phi}_{\mathcal{D}_t(i)}^\top \mathbf{\Phi}_{\mathcal{D}_t(i)} + \lambda \mathbf{I})^{-1/2} \preceq \epsilon_{t,i} \mathbf{I} \\
\Leftrightarrow& \|(\mathbf{\Phi}_{\mathcal{D}_t(i)}^\top \mathbf{\Phi}_{\mathcal{D}_t(i)} + \lambda \mathbf{I})^{-1/2} (\mathbf{\Phi}_{\mathcal{D}_t(i)}^\top \bar{\mathbf{S}}_{t,i}^\top \bar{\mathbf{S}}_{t,i} \mathbf{\Phi}_{\mathcal{D}_t(i)} - \mathbf{\Phi}_{\mathcal{D}_t(i)}^\top \mathbf{\Phi}_{\mathcal{D}_t(i)})(\mathbf{\Phi}_{\mathcal{D}_t(i)}^\top \mathbf{\Phi}_{\mathcal{D}_t(i)} + \lambda \mathbf{I})^{-1/2}\| \leq \epsilon_{t,i} \\
\Leftrightarrow& \| \sum_{s \in \mathcal{D}_{t_p}} (\frac{q_s}{\tilde{p}_s} - 1) \psi_s \psi_s^\top + \sum_{s \in \mathcal{D}_t(i) \backslash \mathcal{D}_{t_p}} (0 - 1) \psi_s \psi_s^\top \| \leq \epsilon_{t,i}
\end{aligned}
$$

where $\psi_s = (\mathbf{\Phi}_{\mathcal{D}_t(i)}^\top \mathbf{\Phi}_{\mathcal{D}_t(i)} + \lambda \mathbf{I})^{-1/2} \phi_s$. Notice that the second term in the norm has weight $-1$ because the dictionary $\mathcal{S}_{t_p}$ is fixed after $t_p$. With triangle inequality, now it suffices to bound

$$
\| \sum_{s \in \mathcal{D}_{t_p}} (\frac{q_s}{\tilde{p}_s} - 1) \psi_{s,j} \psi_s^\top + \sum_{s \in \mathcal{D}_t(i) \backslash \mathcal{D}_{t_p}} (0 - 1) \psi_s \psi_s^\top \| \leq \| \sum_{s \in \mathcal{D}_{t_p}} (\frac{q_s}{\tilde{p}_s} - 1) \psi_s \psi_s^\top \| + \| \sum_{s \in \mathcal{D}_t(i) \backslash \mathcal{D}_{t_p}} \psi_s \psi_s^\top \|.
$$

We should note that the first term corresponds to the approximation accuracy of $\mathcal{S}_{t_p}$ w.r.t. the dataset $\mathcal{D}_{t_p}$. And under the condition that it is $\epsilon$-accurate w.r.t. $\mathcal{D}_{t_p}$, we have $\|\sum_{s \in \mathcal{D}_{t_p}} (\frac{q_s}{\tilde{p}_s} - 1) \psi_s \psi_s^\top\| \leq \epsilon$. The second term measures the difference between $\mathcal{D}_t(i)$ compared with $\mathcal{D}_{t_p}$, which is unique to our

work. We can bound it as follows.

$$\| \sum_{s\in\mathcal{D}_t(i)\backslash\mathcal{D}_{t_p}} \psi_s\psi_s^\top \|$$

$$=\|(\boldsymbol{\Phi}_{\mathcal{D}_t(i)}^\top\boldsymbol{\Phi}_{\mathcal{D}_t(i)}+\lambda\mathbf{I})^{-1/2}(\sum_{s\in\mathcal{D}_t(i)\backslash\mathcal{D}_{t_p}}\phi_s\phi_s^\top)(\boldsymbol{\Phi}_{\mathcal{D}_t(i)}^\top\boldsymbol{\Phi}_{\mathcal{D}_t(i)}+\lambda\mathbf{I})^{-1/2}\|$$

$$=\max_{\phi\in\mathcal{H}}\frac{\phi^\top(\boldsymbol{\Phi}_{\mathcal{D}_t(i)}^\top\boldsymbol{\Phi}_{\mathcal{D}_t(i)}+\lambda\mathbf{I})^{-1/2}(\sum_{s\in\mathcal{D}_t(i)\backslash\mathcal{D}_{t_p}}\phi_s\phi_s^\top)(\boldsymbol{\Phi}_{\mathcal{D}_t(i)}^\top\boldsymbol{\Phi}_{\mathcal{D}_t(i)}+\lambda\mathbf{I})^{-1/2}\phi}{\phi^\top\phi}$$

$$=\max_{\phi\in\mathcal{H}}\frac{\phi^\top(\sum_{s\in\mathcal{D}_t(i)\backslash\mathcal{D}_{t_p}}\phi_s\phi_s^\top)\phi}{\phi^\top(\boldsymbol{\Phi}_{\mathcal{D}_t(i)}^\top\boldsymbol{\Phi}_{\mathcal{D}_t(i)}+\lambda\mathbf{I})\phi}$$

$$=1-\min_{\phi\in\mathcal{H}}\frac{\phi^\top(\boldsymbol{\Phi}_{\mathcal{D}_{t_p}}^\top\boldsymbol{\Phi}_{\mathcal{D}_{t_p}}+\lambda\mathbf{I})\phi}{\phi^\top(\boldsymbol{\Phi}_{\mathcal{D}_t(i)}^\top\boldsymbol{\Phi}_{\mathcal{D}_t(i)}+\lambda\mathbf{I})\phi}$$

$$=1-\frac{1}{\max_{\phi\in\mathcal{H}}\frac{\phi^\top(\boldsymbol{\Phi}_{\mathcal{D}_{t_p}}^\top\boldsymbol{\Phi}_{\mathcal{D}_{t_p}}+\lambda\mathbf{I})^{-1}\phi}{\phi^\top(\boldsymbol{\Phi}_{\mathcal{D}_t(i)}^\top\boldsymbol{\Phi}_{\mathcal{D}_t(i)}+\lambda\mathbf{I})^{-1}\phi}}$$

$$=1-\frac{1}{\max_{\mathbf{x}}\frac{\sigma_{t_p,i}^2(\mathbf{x})}{\sigma_{t,i}^2(\mathbf{x})}}$$

We can further bound the term $\frac{\sigma_{t_p,i}^2(\mathbf{x})}{\sigma_{t,i}^2(\mathbf{x})}$ using the threshold of the event-trigger in Eq (4). For any $\mathbf{x}\in\mathbb{R}^d$,

$$\frac{\sigma_{t_p,i}^2(\mathbf{x})}{\sigma_{t,i}^2(\mathbf{x})}\leq 1+\sum_{s\in\mathcal{D}_t(i)\backslash\mathcal{D}_{t_p}}\hat{\sigma}_{t_p,i}^2(\mathbf{x}_s)\leq 1+\frac{1+\epsilon}{1-\epsilon}\sum_{s\in\mathcal{D}_t(i)\backslash\mathcal{D}_{t_p}}\tilde{\sigma}_{t_p,i}^2(\mathbf{x}_s)\leq 1+\frac{1+\epsilon}{1-\epsilon}D$$

where the first inequality is due to Lemma A.4, the second is due to Lemma E.1, and the third is due to the event-trigger in Eq (4). Putting everything together, we have that if $\mathcal{S}_{t_p}$ is $\epsilon$-accurate w.r.t. $\mathcal{D}_{t_p}$, then it is $\left(\epsilon+1-\frac{1}{1+\frac{1+\epsilon}{1-\epsilon}D}\right)$-accurate w.r.t. dataset $\mathcal{D}_t(i)$, which finishes the proof. $\qquad\square$

## F Proof of Lemma 4.2

To prove Lemma 4.2, we need the following lemma.

**Lemma F.1.** *We have $\forall t,i$ that*

$$\|\tilde{\theta}_{t,i}-\theta_\star\|_{\tilde{\mathbf{A}}_{t,i}}\leq\left(\|\boldsymbol{\Phi}_{\mathcal{D}_t(i)}(\mathbf{I}-\mathbf{P}_\mathcal{S})\|+\sqrt{\lambda}\right)\|\theta_\star\|+R\sqrt{4\ln N/\delta+2\ln\det((1+\lambda)\mathbf{I}+\mathbf{K}_{\mathcal{D}_t(i),\mathcal{D}_t(i)})}$$

*with probability at least $1-\delta$.*

*Proof of Lemma F.1.* Recall that the approximated kernel Ridge regression estimator for $\theta_\star$ is defined as

$$\tilde{\theta}_{t,i}=\tilde{\mathbf{A}}_{t,i}^{-1}\mathbf{P}_\mathcal{S}\boldsymbol{\Phi}_{\mathcal{D}_t(i)}^\top\mathbf{y}_{\mathcal{D}_t(i)}$$

where $\mathbf{P}_\mathcal{S}$ is the orthogonal projection matrix for the Nyström approximation, and $\tilde{\mathbf{A}}_{t,i}=\mathbf{P}_\mathcal{S}\boldsymbol{\Phi}_{\mathcal{D}_t(i)}^\top\boldsymbol{\Phi}_{\mathcal{D}_t(i)}\mathbf{P}_\mathcal{S}+\lambda\mathbf{I}$. Then our goal is to bound

$$(\tilde{\theta}_{t,i}-\theta_\star)^\top\tilde{\mathbf{A}}_{t,i}(\tilde{\theta}_{t,i}-\theta_\star)$$

$$=(\tilde{\theta}_{t,i}-\theta_\star)^\top\tilde{\mathbf{A}}_{t,i}(\tilde{\mathbf{A}}_{t,i}^{-1}\mathbf{P}_\mathcal{S}\boldsymbol{\Phi}_{\mathcal{D}_t(i)}^\top\mathbf{y}_{\mathcal{D}_t(i)}-\theta_\star)$$

$$=(\tilde{\theta}_{t,i}-\theta_\star)^\top\tilde{\mathbf{A}}_{t,i}[\tilde{\mathbf{A}}_{t,i}^{-1}\mathbf{P}_\mathcal{S}\boldsymbol{\Phi}_{\mathcal{D}_t(i)}^\top(\boldsymbol{\Phi}_{\mathcal{D}_t(i)}\theta_\star+\eta_{\mathcal{D}_t(i)})-\theta_\star]$$

$$=(\tilde{\theta}_{t,i}-\theta_\star)^\top\tilde{\mathbf{A}}_{t,i}(\tilde{\mathbf{A}}_{t,i}^{-1}\mathbf{P}_\mathcal{S}\boldsymbol{\Phi}_{\mathcal{D}_t(i)}^\top\boldsymbol{\Phi}_{\mathcal{D}_t(i)}\theta_\star-\theta_\star)+(\tilde{\theta}_{t,i}-\theta_\star)^\top\tilde{\mathbf{A}}_{t,i}\tilde{\mathbf{A}}_{t,i}^{-1}\mathbf{P}_\mathcal{S}\boldsymbol{\Phi}_{\mathcal{D}_t(i)}^\top\eta_{\mathcal{D}_t(i)}$$

**Bounding the first term**  To bound the first term, we begin with rewriting

$$\tilde{\mathbf{A}}_{t,i}(\tilde{\mathbf{A}}_{t,i}^{-1}\mathbf{P}_{\mathcal{S}}\mathbf{\Phi}_{\mathcal{D}_t(i)}^{\top}\mathbf{\Phi}_{\mathcal{D}_t(i)}\theta_\star - \theta_\star)$$
$$=\mathbf{P}_{\mathcal{S}}\mathbf{\Phi}_{\mathcal{D}_t(i)}^{\top}\mathbf{\Phi}_{\mathcal{D}_t(i)}\theta_\star - \mathbf{P}_{\mathcal{S}}\mathbf{\Phi}_{\mathcal{D}_t(i)}^{\top}\mathbf{\Phi}_{\mathcal{D}_t(i)}\mathbf{P}_{\mathcal{S}}\theta_\star - \lambda\theta_\star$$
$$=\mathbf{P}_{\mathcal{S}}\mathbf{\Phi}_{\mathcal{D}_t(i)}^{\top}\mathbf{\Phi}_{\mathcal{D}_t(i)}(\mathbf{I}-\mathbf{P}_{\mathcal{S}})\theta_\star - \lambda\theta_\star$$

and by substituting this into the first term, we have

$$(\tilde{\theta}_{t,i} - \theta_\star)^{\top}\tilde{\mathbf{A}}_{t,i}(\tilde{\mathbf{A}}_{t,i}^{-1}\mathbf{P}_{\mathcal{S}}\mathbf{\Phi}_{\mathcal{D}_t(i)}^{\top}\mathbf{\Phi}_{\mathcal{D}_t(i)}\theta_\star - \theta_\star)$$
$$=(\tilde{\theta}_{t,i} - \theta_\star)^{\top}\mathbf{P}_{\mathcal{S}}\mathbf{\Phi}_{\mathcal{D}_t(i)}^{\top}\mathbf{\Phi}_{\mathcal{D}_t(i)}(\mathbf{I}-\mathbf{P}_{\mathcal{S}})\theta_\star - \lambda(\tilde{\theta}_{t,i} - \theta_\star)^{\top}\theta_\star$$
$$=(\tilde{\theta}_{t,i} - \theta_\star)^{\top}\tilde{\mathbf{A}}_{t,i}^{1/2}\tilde{\mathbf{A}}_{t,i}^{-1/2}\mathbf{P}_{\mathcal{S}}\mathbf{\Phi}_{\mathcal{D}_t(i)}^{\top}\mathbf{\Phi}_{\mathcal{D}_t(i)}(\mathbf{I}-\mathbf{P}_{\mathcal{S}})\theta_\star - \lambda(\tilde{\theta}_{t,i} - \theta_\star)^{\top}\tilde{\mathbf{A}}_{t,i}^{1/2}\tilde{\mathbf{A}}_{t,i}^{-1/2}\theta_\star$$
$$\leq\|\tilde{\theta}_{t,i} - \theta_\star\|_{\tilde{\mathbf{A}}_{t,i}}\big(\|\tilde{\mathbf{A}}_{t,i}^{-1/2}\mathbf{P}_{\mathcal{S}}\mathbf{\Phi}_{\mathcal{D}_t(i)}^{\top}\mathbf{\Phi}_{\mathcal{D}_t(i)}(\mathbf{I}-\mathbf{P}_{\mathcal{S}})\theta_\star\| + \lambda\|\theta_\star\|_{\tilde{\mathbf{A}}_{t,i}^{-1}}\big)$$
$$\leq\|\tilde{\theta}_{t,i} - \theta_\star\|_{\tilde{\mathbf{A}}_{t,i}}\big(\|\tilde{\mathbf{A}}_{t,i}^{-1/2}\mathbf{P}_{\mathcal{S}}\mathbf{\Phi}_{\mathcal{D}_t(i)}^{\top}\|\|\mathbf{\Phi}_{\mathcal{D}_t(i)}(\mathbf{I}-\mathbf{P}_{\mathcal{S}})\|\|\theta_\star\| + \sqrt{\lambda}\|\theta_\star\|\big)$$
$$\leq\|\tilde{\theta}_{t,i} - \theta_\star\|_{\tilde{\mathbf{A}}_{t,i}}\big(\|\mathbf{\Phi}_{\mathcal{D}_t(i)}(\mathbf{I}-\mathbf{P}_{\mathcal{S}})\| + \sqrt{\lambda}\big)\|\theta_\star\|$$

where the first inequality is due to Cauchy Schwartz, and the last inequality is because $\|\tilde{\mathbf{A}}_{t,i}^{-1/2}\mathbf{P}_{\mathcal{S}}\mathbf{\Phi}_{\mathcal{D}_t(i)}^{\top}\| = \sqrt{\mathbf{\Phi}_{\mathcal{D}_t(i)}\mathbf{P}_{\mathcal{S}}(\mathbf{P}_{\mathcal{S}}\mathbf{\Phi}_{\mathcal{D}_t(i)}^{\top}\mathbf{\Phi}_{\mathcal{D}_t(i)}\mathbf{P}_{\mathcal{S}} + \lambda\mathbf{I})^{-1}\mathbf{P}_{\mathcal{S}}\mathbf{\Phi}_{\mathcal{D}_t(i)}^{\top}} \leq 1$.

**Bounding the second term**  By applying Cauchy-Schwartz inequality to the second term, we have

$$(\tilde{\theta}_{t,i} - \theta_\star)^{\top}\tilde{\mathbf{A}}_{t,i}\tilde{\mathbf{A}}_{t,i}^{-1}\mathbf{P}_{\mathcal{S}}\mathbf{\Phi}_{\mathcal{D}_t(i)}^{\top}\eta_{\mathcal{D}_t(i)}$$
$$\leq\|\tilde{\theta}_{t,i} - \theta_\star\|_{\tilde{\mathbf{A}}_{t,i}}\|\tilde{\mathbf{A}}_{t,i}^{-1/2}\mathbf{P}_{\mathcal{S}}\mathbf{\Phi}_{\mathcal{D}_t(i)}^{\top}\eta_{\mathcal{D}_t(i)}\|$$
$$=\|\tilde{\theta}_{t,i} - \theta_\star\|_{\tilde{\mathbf{A}}_{t,i}}\|\tilde{\mathbf{A}}_{t,i}^{-1/2}\mathbf{P}_{\mathcal{S}}\mathbf{A}_{t,i}^{1/2}\mathbf{A}_{t,i}^{-1/2}\mathbf{\Phi}_{\mathcal{D}_t(i)}^{\top}\eta_{\mathcal{D}_t(i)}\|$$
$$\leq\|\tilde{\theta}_{t,i} - \theta_\star\|_{\tilde{\mathbf{A}}_{t,i}}\|\tilde{\mathbf{A}}_{t,i}^{-1/2}\mathbf{P}_{\mathcal{S}}\mathbf{A}_{t,i}^{1/2}\|\|\mathbf{A}_{t,i}^{-1/2}\mathbf{\Phi}_{\mathcal{D}_t(i)}^{\top}\eta_{\mathcal{D}_t(i)}\|$$

Note that $\mathbf{P}_{\mathcal{S}}\mathbf{A}_{t,i}\mathbf{P}_{\mathcal{S}} = \mathbf{P}_{\mathcal{S}}(\mathbf{\Phi}_{\mathcal{D}_t(i)}^{\top}\mathbf{\Phi}_{\mathcal{D}_t(i)} + \lambda\mathbf{I})\mathbf{P}_{\mathcal{S}} = \tilde{\mathbf{A}}_{t,i} + \lambda(\mathbf{P}_{\mathcal{S}} - \mathbf{I})$ and $\mathbf{P}_{\mathcal{S}} \preceq \mathbf{I}$, so we have

$$\|\tilde{\mathbf{A}}_{t,i}^{-1/2}\mathbf{P}_{\mathcal{S}}\mathbf{A}_{t,i}^{1/2}\| = \sqrt{\|\tilde{\mathbf{A}}_{t,i}^{-1/2}\mathbf{P}_{\mathcal{S}}\mathbf{A}_{t,i}^{1/2}\mathbf{A}_{t,i}^{1/2}\mathbf{P}_{\mathcal{S}}\tilde{\mathbf{A}}_{t,i}^{-1/2}\|} \leq \sqrt{\|\tilde{\mathbf{A}}_{t,i}^{-1/2}(\tilde{\mathbf{A}}_{t,i} + \lambda(\mathbf{P}_{\mathcal{S}} - \mathbf{I}))\tilde{\mathbf{A}}_{t,i}^{-1/2}\|}$$
$$= \sqrt{\|\mathbf{I} + \lambda\tilde{\mathbf{A}}_{t,i}^{-1/2}(\mathbf{P}_{\mathcal{S}} - \mathbf{I}))\tilde{\mathbf{A}}_{t,i}^{-1/2}\|} \leq \sqrt{1 + \lambda\|\tilde{\mathbf{A}}_{t,i}^{-1}\|\|\mathbf{P}_{\mathcal{S}} - \mathbf{I}\|}$$
$$\leq \sqrt{1 + \lambda \cdot \lambda^{-1} \cdot 1} = \sqrt{2}$$

Then using the self-normalized bound derived for Lemma B.1, the term $\|\mathbf{A}_{t,i}^{-1/2}\mathbf{\Phi}_{\mathcal{D}_t(i)}^{\top}\eta_{\mathcal{D}_t(i)}\| = \|\mathbf{\Phi}_{\mathcal{D}_t(i)}^{\top}\eta_{\mathcal{D}_t(i)}\|_{\mathbf{A}_{t,i}^{-1}}$ can be bounded by

$$\|\mathbf{\Phi}_{\mathcal{D}_t(i)}^{\top}\eta_{\mathcal{D}_t(i)}\|_{\mathbf{A}_{t,i}^{-1}} \leq R\sqrt{2\ln(NT/\delta) + \ln(\det(\mathbf{K}_{\mathcal{D}_t(i),\mathcal{D}_t(i)}/\lambda + \mathbf{I}))}$$
$$\leq R\sqrt{2\ln(NT/\delta) + 2\gamma_{NT}}$$

for $\forall t, i$, with probability at least $1 - \delta$. Combining everything finishes the proof.  $\square$

Now we are ready to prove Lemma 4.2 by further bounding the term $\|\mathbf{\Phi}_{\mathcal{D}_t(i)}(\mathbf{I} - \mathbf{P}_{\mathcal{S}_{t_p}})\|$.

*Proof of Lemma 4.2.*  Recall that $\bar{\mathbf{S}}_{t,i} \in \mathbb{R}^{|\mathcal{D}_t(i)|\times|\mathcal{D}_t(i)|}$ denotes the diagonal matrix, whose $s$-th diagonal entry equals to $\frac{q_s}{\sqrt{\tilde{p}_s}}$, where $q_s = 1$ if $s \in \mathcal{S}_{t_p}$ and 0 otherwise (note that for $s \notin \mathcal{S}_{t_p}$, we set $\tilde{p}_s = 1$, so $q_s/\tilde{p}_s = 0$). Therefore, $\forall s \in \mathcal{D}_t(i) \setminus \mathcal{D}_{t_p}, q_s = 0$, as the dictionary is fixed after $t_p$. We can rewrite $\mathbf{\Phi}_{\mathcal{D}_t(i)}^{\top}\bar{\mathbf{S}}_{t,i}^{\top}\bar{\mathbf{S}}_{t,i}\mathbf{\Phi}_{\mathcal{D}_t(i)} = \sum_{s\in\mathcal{D}_t(i)}\frac{q_s}{\tilde{p}_s}\phi_s\phi_s^{\top}$, where $\phi_s := \phi(\mathbf{x}_s)$. Then by definition of the spectral norm $\|\cdot\|$, and the properties of the projection matrix $\mathbf{P}_{\mathcal{S}_{t_p}}$, we have

$$\|\mathbf{\Phi}_{\mathcal{D}_t(i)}(\mathbf{I} - \mathbf{P}_{\mathcal{S}_{t_p}})\| = \sqrt{\lambda_{\max}\big(\mathbf{\Phi}_{\mathcal{D}_t(i)}(\mathbf{I} - \mathbf{P}_{\mathcal{S}_{t_p}})^2\mathbf{\Phi}_{\mathcal{D}_t(i)}^{\top}\big)} = \sqrt{\lambda_{\max}\big(\mathbf{\Phi}_{\mathcal{D}_t(i)}(\mathbf{I} - \mathbf{P}_{\mathcal{S}_{t_p}})\mathbf{\Phi}_{\mathcal{D}_t(i)}^{\top}\big)}.$$
$$(6)$$

Moreover, due to Lemma E.2, we know $\mathcal{S}_{t_p}$ is $\epsilon_{t,i}$-accurate w.r.t. $\mathcal{D}_t(i)$ for $t \in [t_p + 1, t_{p+1}]$, where $\epsilon_{t,i} = \left(\epsilon + 1 - \frac{1}{1+\frac{1+\epsilon}{1-\epsilon}D}\right)$, so we have $\mathbf{I} - \mathbf{P}_{\mathcal{S}_{t_p}} \preceq \frac{\lambda}{1-\epsilon_{t,i}}(\mathbf{\Phi}_{\mathcal{D}_t(i)}^\top \mathbf{\Phi}_{\mathcal{D}_t(i)} + \lambda\mathbf{I})^{-1}$ by the property of $\epsilon$-accuracy (Proposition 10 of [3]). Therefore, by substituting this back to Eq (6), we have

$$\|\mathbf{\Phi}_{\mathcal{D}_t(i)}(\mathbf{I} - \mathbf{P}_{\mathcal{S}_{t_p}})\| \leq \sqrt{\lambda_{\max}\left(\frac{\lambda}{1-\epsilon_{t,i}}\mathbf{\Phi}_{\mathcal{D}_t(i)}(\mathbf{\Phi}_{\mathcal{D}_t(i)}^\top \mathbf{\Phi}_{\mathcal{D}_t(i)} + \lambda\mathbf{I})^{-1}\mathbf{\Phi}_{\mathcal{D}_t(i)}^\top\right)} \leq \sqrt{\frac{\lambda}{-\epsilon + \frac{1}{1+\frac{1+\epsilon}{1-\epsilon}D}}}$$

which finishes the proof. $\qquad\square$

## G  Proof of Theorem 4.3: Regret and Communication Cost of Approx-DisKernelUCB

### G.1  Regret Analysis

Consider some time step $t \in [t_{p-1} + 1, t_p]$, where $p \in [B]$. Due to Lemma 4.2, i.e., the confidence ellipsoid for approximated estimator, and the fact that $\mathbf{x}_t = \arg\max_{\mathbf{x} \in \mathcal{A}_{t,i}} \tilde{\mu}_{t-1,i}(\mathbf{x}) + \alpha_{t-1,i}\tilde{\sigma}_{t-1,i}(\mathbf{x})$, we have

$$f(\mathbf{x}_t^\star) \leq \tilde{\mu}_{t-1,i}(\mathbf{x}_t^\star) + \alpha_{t-1,i}\tilde{\sigma}_{t-1,i}(\mathbf{x}_t^\star) \leq \tilde{\mu}_{t-1,i}(\mathbf{x}_t) + \alpha_{t-1,i}\tilde{\sigma}_{t-1,i}(\mathbf{x}_t),$$
$$f(\mathbf{x}_t) \geq \tilde{\mu}_{t-1,i}(\mathbf{x}_t) - \alpha_{t-1,i}\tilde{\sigma}_{t-1,i}(\mathbf{x}_t),$$

and thus $r_t = f(\mathbf{x}_t^\star) - f(\mathbf{x}_t) \leq 2\alpha_{t-1,i}\tilde{\sigma}_{t-1,i}(\mathbf{x}_t)$, where

$$\alpha_{t-1,i} = \left(\frac{1}{\sqrt{-\epsilon + \frac{1}{1+\frac{1+\epsilon}{1-\epsilon}D}}} + 1\right)\sqrt{\lambda}\|\theta_\star\| + R\sqrt{4\ln NT/\delta + 2\ln\det((1+\lambda)\mathbf{I} + \mathbf{K}_{\mathcal{D}_{t-1}(i),\mathcal{D}_{t-1}(i)})}.$$

Note that, different from Appendix C, the $\alpha_{t-1,i}$ term now depends on the threshold $D$ and accuracy constant $\epsilon$, as a result of the approximation error. As we will see in the following paragraphs, their values need to be set properly in order to bound $\alpha_{t-1,i}$.

We begin the regret analysis of Approx-DisKernelUCB with the same decomposition of good and bad epochs as in Appendix C.1, i.e., we call the $p$-th epoch a good epoch if $\ln\left(\frac{\det(\mathbf{I}+\lambda^{-1}\mathbf{K}_{[t_p],[t_p]})}{\det(\mathbf{I}+\lambda^{-1}\mathbf{K}_{[t_{p-1}],[t_{p-1}]})}\right) \leq 1$, otherwise it is a bad epoch. Moreover, due to the pigeon-hold principle, there can be at most $2\gamma_{NT}$ bad epochs.

As we will show in the following paragraphs, using Lemma E.1, we can obtain a similar bound for the cumulative regret in good epochs as that in Appendix C.1, but with additional dependence on $D$ and $\epsilon$. The proof mainly differs in the bad epochs, where we need to use the event-trigger in Eq (4) to bound the cumulative regret in each bad epoch. Compared with Eq (2), Eq (4) does not contain the number of local updates on each client since last synchronization., and as mentioned in Section 4.2, this introduces a $\sqrt{T}$ factor in the regret bound for bad epochs in place of the $\sqrt{\gamma_{NT}}$ term in Appendix C.1.

**Cumulative Regret in Good Epochs**   Let's first consider some time step $t$ in a good epoch $p$, i.e., $t \in [t_{p-1} + 1, t_p]$, and we have the following bound on the instantaneous regret

$$r_t \leq 2\alpha_{t-1,i}\tilde{\sigma}_{t-1,i}(\mathbf{x}_t) \leq 2\alpha_{t-1,i}\tilde{\sigma}_{t_{p-1},i}(\mathbf{x}_t) \leq 2\alpha_{t-1,i}\frac{1+\epsilon}{1-\epsilon}\sigma_{t_{p-1},i}(\mathbf{x}_t)$$

$$= 2\alpha_{t-1,i}\frac{1+\epsilon}{1-\epsilon}\sqrt{\phi_t^\top A_{t_{p-1}}^{-1}\phi_t} \leq 2\alpha_{t-1,i}\frac{1+\epsilon}{1-\epsilon}\sqrt{\phi_t^\top A_{t-1}^{-1}\phi_t}\sqrt{\frac{\det(\mathbf{I}+\lambda^{-1}\mathbf{K}_{[t-1],[t-1]})}{\det(\mathbf{I}+\lambda^{-1}\mathbf{K}_{[t_{p-1}],[t_{p-1}]})}}$$

$$\leq 2\sqrt{e}\frac{1+\epsilon}{1-\epsilon}\alpha_{t-1,i}\sqrt{\phi_t^\top A_{t-1}^{-1}\phi_t}$$

where the second inequality is because the (approximated) variance is non-decreasing, the third inequality is due to Lemma E.1, the forth is due to Lemma A.2, and the last is because in a good epoch, we have $\frac{\det(\mathbf{I}+\lambda^{-1}\mathbf{K}_{[t-1],[t-1]})}{\det(\mathbf{I}+\lambda^{-1}\mathbf{K}_{[t_{p-1}],[t_{p-1}]})} \leq \frac{\det(\mathbf{I}+\lambda^{-1}\mathbf{K}_{[t_p],[t_p]})}{\det(\mathbf{I}+\lambda^{-1}\mathbf{K}_{[t_{p-1}],[t_{p-1}]})} \leq e$ for $t \in [t_{p-1} + 1, t_p]$.

Therefore, the cumulative regret incurred in all good epochs, denoted by $R_{good}$, is upper bounded by

$$R_{good} \le 2\sqrt{e}\frac{1+\epsilon}{1-\epsilon}\sum_{t=1}^{NT}\alpha_{t-1,i}\sqrt{\phi_t^\top A_{t-1}^{-1}\phi_t} \le 2\sqrt{e}\frac{1+\epsilon}{1-\epsilon}\alpha_{NT}\sqrt{NT\cdot\sum_{t=1}^{NT}\phi_t^\top A_{t-1}^{-1}\phi_t}$$

$$\le 2\sqrt{e}\frac{1+\epsilon}{1-\epsilon}\alpha_{NT}\sqrt{NT\cdot 2\gamma_{NT}}$$

where $\alpha_{NT} := \left(\frac{1}{\sqrt{-\epsilon+\frac{1}{1+\frac{1+\epsilon}{1-\epsilon}D}}}+1\right)\sqrt{\lambda}\|\theta_\star\|+R\sqrt{4\ln NT/\delta+2\ln\det((1+\lambda)\mathbf{I}+\mathbf{K}_{[NT],[NT]})}$.

**Cumulative Regret in Bad Epochs**   The cumulative regret incurred in this bad epoch is

$$\sum_{p=1}^{B}\mathbb{1}\{\ln(\frac{\det(\mathbf{I}+\lambda^{-1}\mathbf{K}_{[t_p],[t_p]})}{\det(\mathbf{I}+\lambda^{-1}\mathbf{K}_{[t_{p-1}],[t_{p-1}]})})>1\}\sum_{t=t_{p-1}+1}^{t_p}r_t$$

$$\le 2\sum_{p=1}^{B}\mathbb{1}\{\ln(\frac{\det(\mathbf{I}+\lambda^{-1}\mathbf{K}_{[t_p],[t_p]})}{\det(\mathbf{I}+\lambda^{-1}\mathbf{K}_{[t_{p-1}],[t_{p-1}]})})>1\}\sum_{t=t_{p-1}+1}^{t_p}\alpha_{t-1,i}\tilde\sigma_{t-1,i}(\mathbf{x}_t)$$

$$\le 2\alpha_{NT}\sum_{p=1}^{B}\mathbb{1}\{\ln(\frac{\det(\mathbf{I}+\lambda^{-1}\mathbf{K}_{[t_p],[t_p]})}{\det(\mathbf{I}+\lambda^{-1}\mathbf{K}_{[t_{p-1}],[t_{p-1}]})})>1\}\sum_{i=1}^{N}\sum_{t\in\mathcal{N}_{t_p}(i)\setminus\mathcal{N}_{t_{p-1}}(i)}\tilde\sigma_{t-1,i}(\mathbf{x}_t)$$

$$\le 2\alpha_{NT}\sum_{p=1}^{B}\mathbb{1}\{\ln(\frac{\det(\mathbf{I}+\lambda^{-1}\mathbf{K}_{[t_p],[t_p]})}{\det(\mathbf{I}+\lambda^{-1}\mathbf{K}_{[t_{p-1}],[t_{p-1}]})})>1\}\sum_{i=1}^{N}\sqrt{(|\mathcal{N}_{t_p}(i)|-|\mathcal{N}_{t_{p-1}}(i)|)\sum_{t\in\mathcal{N}_{t_p}(i)\setminus\mathcal{N}_{t_{p-1}}(i)}\tilde\sigma_{t-1,i}^2(\mathbf{x}_t)}$$

$$\le 2\alpha_{NT}\sqrt{D}\sum_{p=1}^{B}\mathbb{1}\{\ln(\frac{\det(\mathbf{I}+\lambda^{-1}\mathbf{K}_{[t_p],[t_p]})}{\det(\mathbf{I}+\lambda^{-1}\mathbf{K}_{[t_{p-1}],[t_{p-1}]})})>1\}\sum_{i=1}^{N}\sqrt{(|\mathcal{N}_{t_p}(i)|-|\mathcal{N}_{t_{p-1}}(i)|)}$$

$$\le 2\alpha_{NT}\sqrt{D}\sum_{p=1}^{B}\mathbb{1}\{\ln(\frac{\det(\mathbf{I}+\lambda^{-1}\mathbf{K}_{[t_p],[t_p]})}{\det(\mathbf{I}+\lambda^{-1}\mathbf{K}_{[t_{p-1}],[t_{p-1}]})})>1\}\sum_{i=1}^{N}\sqrt{\frac{t_p-t_{p-1}}{N}}$$

$$\le 2\alpha_{NT}\sqrt{DN}\sum_{p=1}^{B}\mathbb{1}\{\ln(\frac{\det(\mathbf{I}+\lambda^{-1}\mathbf{K}_{[t_p],[t_p]})}{\det(\mathbf{I}+\lambda^{-1}\mathbf{K}_{[t_{p-1}],[t_{p-1}]})})>1\}\sqrt{t_p-t_{p-1}}$$

$$\le 2\alpha_{NT}\sqrt{DN}\sqrt{\sum_{p=1}^{B}\mathbb{1}\{\ln(\frac{\det(\mathbf{I}+\lambda^{-1}\mathbf{K}_{[t_p],[t_p]})}{\det(\mathbf{I}+\lambda^{-1}\mathbf{K}_{[t_{p-1}],[t_{p-1}]})})>1\}(t_p-t_{p-1})\cdot\sum_{p=1}^{B}\mathbb{1}\{\ln(\frac{\det(\mathbf{I}+\lambda^{-1}\mathbf{K}_{[t_p],[t_p]})}{\det(\mathbf{I}+\lambda^{-1}\mathbf{K}_{[t_{p-1}],[t_{p-1}]})})>1\}}$$

$$\le 2\alpha_{NT}\sqrt{DN}\sqrt{2NT\gamma_{NT}}$$

where the third inequality is due to the Cauchy-Schwartz inequality, the forth is due to our event-trigger in Eq (4), the fifth is due to our assumption that clients interact with the environment in a round-robin manner, the sixth is due to the Cauchy-Schwartz inequality again, and the last is due to the fact that there can be at most $2\gamma_{NT}$ bad epochs.

Combining cumulative regret incurred during both good and bad epochs, we have

$$R_{NT}\le R_{good}+R_{bad}\le 2\sqrt{e}\frac{1+\epsilon}{1-\epsilon}\alpha_{NT}\sqrt{NT\cdot 2\gamma_{NT}}+2\alpha_{NT}\sqrt{DN}\sqrt{2NT\gamma_{NT}}$$

### G.2   Communication Cost Analysis

Consider some epoch $p$. We know that for the client $i$ who triggers the global synchronization, we have

$$\frac{1+\epsilon}{1-\epsilon}\sum_{s=t_{p-1}+1}^{t_p}\sigma_{t_{p-1}}^2(\mathbf{x}_s)\ge\sum_{s=t_{p-1}+1}^{t_p}\tilde\sigma_{t_{p-1}}^2(\mathbf{x}_s)\ge\sum_{s\in\mathcal{D}_{t_p(i)}\setminus\mathcal{D}_{t_{p-1}(i)}}\tilde\sigma_{t_{p-1}}^2(\mathbf{x}_s)\ge D$$

Then by summing over $B$ epochs, we have

$$BD<\frac{1+\epsilon}{1-\epsilon}\sum_{p=1}^{B}\sum_{s=t_{p-1}+1}^{t_p}\sigma_{t_{p-1}}^2(\mathbf{x}_s)\le\frac{1+\epsilon}{1-\epsilon}\sum_{p=1}^{B}\sum_{s=t_{p-1}+1}^{t_p}\sigma_{s-1}^2(\mathbf{x}_s)\frac{\sigma_{t_{p-1}}^2(\mathbf{x}_s)}{\sigma_{s-1}^2(\mathbf{x}_s)}.$$

Now we need to bound the ratio $\frac{\sigma_{t_{p-1}}^2(\mathbf{x}_s)}{\sigma_{s-1}^2(\mathbf{x}_s)}$ for $s \in [t_{p-1}+1, t_p]$.

$$\frac{\sigma_{t_{p-1}}^2(\mathbf{x}_s)}{\sigma_{s-1}^2(\mathbf{x}_s)} \le \left[1 + \sum_{\tau=t_{p-1}+1}^{s} \sigma_{t_{p-1}}^2(\mathbf{x}_\tau)\right] \le \left[1 + \frac{1+\epsilon}{1-\epsilon}\sum_{\tau=t_{p-1}+1}^{s} \tilde{\sigma}_{t_{p-1}}^2(\mathbf{x}_\tau)\right]$$

Note that for the client who triggers the global synchronization, we have $\sum_{s \in \mathcal{D}_{t_{p-1}}(i) \backslash \mathcal{D}_{t_{p-1}}(i)} \tilde{\sigma}_{t_{p-1}}^2(\mathbf{x}_s) < D$, i.e., one time step before it triggers the synchronization at time $t_p$. Due to the fact that the (approximated) posterior variance cannot exceed $L^2/\lambda$, we have $\sum_{s \in \mathcal{D}_{t_p}(i) \backslash \mathcal{D}_{t_{p-1}}(i)} \tilde{\sigma}_{t_{p-1}}^2(\mathbf{x}_s) < D + L^2/\lambda$. For the other $N-1$ clients, we have $\sum_{s \in \mathcal{D}_{t_p}(i) \backslash \mathcal{D}_{t_{p-1}}(i)} \tilde{\sigma}_{t_{p-1}}^2(\mathbf{x}_s) < D$. Summing them together, we have

$$\sum_{s=t_{p-1}+1}^{t_p} \tilde{\sigma}_{t_{p-1}}^2(\mathbf{x}_s) < (ND + L^2/\lambda)$$

for the $p$-th epoch. By substituting this back, we have

$$\frac{\sigma_{t_{p-1}}^2(\mathbf{x}_s)}{\sigma_{s-1}^2(\mathbf{x}_s)} \le \left[1 + \frac{1+\epsilon}{1-\epsilon}(ND + L^2/\lambda)\right]$$

Therefore,

$$BD < \frac{1+\epsilon}{1-\epsilon}\left[1 + \frac{1+\epsilon}{1-\epsilon}(ND + L^2/\lambda)\right]\sum_{p=1}^{B}\sum_{s=t_{p-1}+1}^{t_p} \sigma_{s-1}^2(\mathbf{x}_s)$$

$$\le \frac{1+\epsilon}{1-\epsilon}\left[1 + \frac{1+\epsilon}{1-\epsilon}(ND + L^2/\lambda)\right]2\gamma_{NT}$$

and thus the total number of epochs $B < \frac{1+\epsilon}{1-\epsilon}[\frac{1}{D} + \frac{1+\epsilon}{1-\epsilon}(N + L^2/(\lambda D))]2\gamma_{NT}$.

By setting $D = \frac{1}{N}$, we have

$$\alpha_{NT} = \left(\frac{1}{\sqrt{-\epsilon + \frac{1}{1 + \frac{1+\epsilon}{1-\epsilon}\frac{1}{N}}}} + 1\right)\sqrt{\lambda}\|\theta_\star\| + R\sqrt{4\ln N/\delta + 2\ln\det((1+\lambda)\mathbf{I} + \mathbf{K}_{[NT],[NT]})}$$

$$\le \left(\frac{1}{\sqrt{-\epsilon + \frac{1}{1 + \frac{1+\epsilon}{1-\epsilon}}}} + 1\right)\sqrt{\lambda}\|\theta_\star\| + R\sqrt{4\ln N/\delta + 2\ln\det((1+\lambda)\mathbf{I} + \mathbf{K}_{[NT],[NT]})}$$

because $N \ge 1$. Moreover, to ensure $-\epsilon + \frac{1}{1+\frac{1+\epsilon}{1-\epsilon}} > 0$, we need to set the constant $\epsilon < 1/3$. Therefore,

$$R_{NT} = O\left(\sqrt{NT}(\|\theta_\star\|\sqrt{\gamma_{NT}} + \gamma_{NT})\right)$$

and the total number of global synchronizations $B = O(N\gamma_{NT})$. Since for each global synchronization, the communication cost is $O(N\gamma_{NT}^2)$, we have

$$C_{NT} = O\left(N^2\gamma_{NT}^3\right)$$

## H  Experiment Setup

**Synthetic dataset**   We simulated the distributed bandit setting defined in Section 3.1, with $d = 20, T = 100, N = 100$ ($NT = 10^4$ interactions in total). In each round $l \in [T]$, each client $i \in [N]$ (denote $t = N(l-1) + i$) selects an arm from candidate set $\mathcal{A}_t$, where $\mathcal{A}_t$ is uniformly sampled from a $\ell_2$ unit ball, with $|\mathcal{A}_t| = 20$. Then the corresponding reward is generated using one of the following reward functions:

$$f_1(\mathbf{x}) = \cos(3\mathbf{x}^\top\theta_\star)$$
$$f_2(\mathbf{x}) = (\mathbf{x}^\top\theta_\star)^3 - 3(\mathbf{x}^\top\theta_\star)^2 - (\mathbf{x}^\top\theta_\star) + 3$$

where the parameter $\theta_\star$ is uniformly sampled from a $\ell_2$ unit ball.

**UCI Datasets**   To evaluate Approx-DisKernelUCB's performance in a more challenging and practical scenario, we performed experiments using real-world datasets: MagicTelescope, Mushroom and Shuttle from the UCI Machine Learning Repository [8]. To convert them to contextual bandit problems, we pre-processed these datasets following the steps in [12]. In particular, we partitioned the dataset in to 20 clusters using k-means, and used the centroid of each cluster as the context vector for the arm and the averaged response variable as mean reward (the response variable is binarized by associating one class as 1, and all the others as 0). Then we simulated the distributed bandit learning problem in Section 3.1 with $|\mathcal{A}_t| = 20$, $T = 100$ and $N = 100$ ($NT = 10^4$ interactions in total).

**MovieLens and Yelp dataset**   Yelp dataset, which is released by the Yelp dataset challenge, consists of 4.7 million rating entries for 157 thousand restaurants by 1.18 million users. MovieLens is a dataset consisting of 25 million ratings between 160 thousand users and 60 thousand movies [13]. Following the pre-processing steps in [2], we built the rating matrix by choosing the top 2000 users and top 10000 restaurants/movies and used singular-value decomposition (SVD) to extract a 10-dimension feature vector for each user and restaurant/movie. We treated rating greater than 2 as positive. We simulated the distributed bandit learning problem in Section 3.1 with $T = 100$ and $N = 100$ ($NT = 10^4$ interactions in total). In each time step, the candidate set $\mathcal{A}_t$ (with $|\mathcal{A}_t| = 20$) is constructed by sampling an arm with reward 1 and nineteen arms with reward 0 from the arm pool, and the concatenation of user and restaurant/movie feature vector is used as the context vector for the arm (thus $d = 20$).

# I   Lower Bound for Distributed Kernelized Contextual Bandits

First, we need the following two lemmas

**Lemma I.1** (Theorem 1 of [29]). *There exists a constant $C > 0$, such that for any instance of kernelized bandit with $L = S = R = 1$, the expected cumulative regret for KernelUCB algorithm is upper bounded by $\mathbf{E}[R_T] \leq C\sqrt{T\gamma_T}$, where the maximum information gain $\gamma_T = O\big((\ln(T))^{d+1}\big)$ for Squared Exponential kernels.*

**Lemma I.2** (Theorem 2 of [23]). *There always exists a set of hard-to-learn instances of kernelized bandit with $L = S = R = 1$, such that for any algorithm, for a uniformly random instance in the set, the expected cumulative regret $\mathbf{E}[R_T] \geq c\sqrt{T(\ln(T))^{d/2}}$ for Squared Exponential kernels, with some constant $c$.*

Then we follow a similar procedure as the proof for Theorem 2 of [30] and Theorem 5.3 of [14], to prove the following lower bound results for distributed kernelized bandit with Squared Exponential kernels.

**Theorem I.3.** *For any distributed kernelized bandit algorithm with expected communication cost less than $O(\frac{N}{(\ln(T))^{0.25d+0.5}})$, there exists a kernelized bandit instance with Squared Exponential kernel, and $L = S = R = 1$, such that the expected cumulative regret for this algorithm is at least $\Omega(N\sqrt{T(\ln(T))^{d/2}})$.*

*Proof of Theorem I.3.*   Here we consider kernelized bandit with Squared Exponential kernels. The proof relies on the construction of a auxiliary algorithm, denoted by **AuxAlg**, based on the original distributed kernelized bandit algorithm, denoted by **DisKernelAlg**, as shown below. For each agent $i \in [N]$, **AuxAlg** performs **DisKernelAlg**, until any communication happens between client $i$ and the server, in which case, **AuxAlg** switches to the single-agent optimal algorithm, i.e., the KernelUCB algorithm that attains the rate in Lemma I.1. Therefore, **AuxAlg** is a single-agent bandit algorithm, and the lower bound in Lemma I.2 applies: the cumulative regret that **AuxAlg** incurs for some agent $i \in [N]$ is lower bounded by

$$\mathbf{E}[R_{\textbf{AuxAlg},i}] \geq c\sqrt{T(\ln(T))^{d/2}},$$

and by summing over all $N$ clients, we have

$$\mathbf{E}[R_{\textbf{AuxAlg}}] = \sum_{i=1}^{N} \mathbf{E}[R_{\textbf{AuxAlg},i}] \geq cN\sqrt{T(\ln(T))^{d/2}}.$$

For each client $i \in [N]$, denote the probability that client $i$ will communicate with the server as $p_i$, and $p := \sum_{i=1}^{N} p_i$. Note that before the communication, the cumulative regret incurred by **AuxAlg** is the same as **DisKernelAlg**, and after the communication happens, the regret incurred by **AuxAlg** is the same as KernelUCB, whose upper bound is given in Lemma I.1. Therefore, the cumulative regret that **AuxAlg** incurs for client $i$ can be upper bounded by

$$\mathbf{E}[R_{\textbf{AuxAlg},i}] \leq \mathbf{E}[R_{\textbf{DisKernelAlg},i}] + p_i C \sqrt{T(\ln(T))^{d+1}},$$

and by summing over $N$ clients, we have

$$\mathbf{E}[R_{\textbf{AuxAlg}}] = \sum_{i=1}^{N} \mathbf{E}[R_{\textbf{AuxAlg},i}]$$
$$\leq \sum_{i=1}^{N} \mathbf{E}[R_{\textbf{DisKernelAlg},i}] + (\sum_{i=1}^{N} p_i) C \sqrt{T(\ln(T))^{d+1}}$$
$$= \mathbf{E}[R_{\textbf{DisKernelAlg}}] + p C \sqrt{T(\ln(T))^{d+1}}.$$

Combining the upper and lower bounds for $\mathbf{E}[R_{\textbf{AuxAlg}}]$, we have

$$\mathbf{E}[R_{\textbf{DisKernelAlg}}] \geq cN \sqrt{T(\ln(T))^{d/2}} - pC \sqrt{T(\ln(T))^{d+1}}.$$

Therefore, for any **DisKernelAlg** with number of communications $p \leq N \frac{c}{2C(\ln(T))^{0.25d+0.5}} = O(\frac{N}{(\ln(T))^{0.25d+0.5}})$, we have

$$\mathbf{E}[R_{\textbf{DisKernelAlg}}] \geq \frac{c}{2} N \sqrt{T(\ln(T))^{d/2}} = \Omega(N \sqrt{T(\ln(T))^{d/2}}).$$

$\square$