# OpenReview forum: "Communication Efficient Distributed Learning for Kernelized Contextual Bandits"
_NeurIPS.cc/2022/Conference — NeurIPS 2022 Accept_

### Official Review · Reviewer_fUXs · 2022-07-10

**Rating:** 5
**Confidence:** 4
**Soundness:** 3 good
**Presentation:** 4 excellent
**Contribution:** 2 fair

**Summary:**

The paper studies the problem of distributed bandits when the reward function lies in an RKHS. The paper proposes algorithms to enable clients to make a decision at each round. The regret and communication cost of proposed algorithms are studied.

**Questions:**

What is the relations between the regret bound of Approx-DisKernelUCB and the Nystrom method to map $x$? How the approximation affect the regret bound? I think it is useful if authors can elaborate more on this in the paper.

**Limitations:**

The paper assumes that the rewards are generated using a kernel function belongs to an RKHS. Furthermore, privacy of clients cannot be protected using the proposed algorithms.

**Strengths And Weaknesses:**

Strength: The paper proposes an algorithm based on Nystrom approximation which can achieve sub-linear regret of $\mathcal{O}(\sqrt{NT})$ which is proved to require communication cost of $\mathcal{O}(N^2)$.

Weaknesses:
1. Clients needs to send their data samples to the server. Therefore, the proposed algorithms cannot protect the privacy of clients.
2. The novelty of the proposed algorithm DisKernelUCB seems to be limited. Since the communication cost of the algorithm should be $\mathcal{O}(TN^2d)$ to achieve regret of $\mathcal{O}(\sqrt{NT})$, it seems that DisKernelUCB is a simple extension of centralized algorithm where each client locally learns to take an action. In fact, it seems that making minor changes in the centralized algorithm can lead to this result.

---

> ### Author Response · Authors · 2022-08-02
> **Response to Reviewer fUXs**
>
> **[Q1]** Clients needs to send their data samples to the server. Therefore, the proposed algorithms cannot protect the privacy of clients.
>
> **[R1]** We want to clarify that the main focus of our paper is to improve the communication efficiency for distributed kernelized bandits, where all participants trust each other and the platform. And thus the risk of privacy leakage is not a main concern of this work. This is a key difference against federated learning, where the transfer of raw data samples is explicitly restricted.
>
> That being said, if we consider privacy as whether data samples are shared as suggested by the reviewer, our Approx-DisKernelUCB already has improved privacy compared with the prior work in distributed kernelized bandits [9] (this is the index of citations in the updated version of our paper), under this less formal privacy definition. [9] needs to transfer all data samples, while our proposed Approx-DisKernelUCB only requires the transfer of a much smaller subset that scales linearly w.r.t. the maximum information gain $\gamma_{NT}$, i.e., a much smaller amount of data samples are at risk.
>
> However, we should note that under a more formal privacy definition like differential privacy, neither our work nor the prior works in distributed linear bandits [19, 30] are private, i.e., simply avoiding sharing raw data is not enough. Instead, we have to introduce random noises to the shared information (e.g., the Gaussian or Laplacian mechanism). In our case, we may need to add randomly generated/fake data samples to the transferred dictionary to ensure privacy, which is an interesting future direction of our work.
>
> **[Q2]** The novelty of the proposed algorithm DisKernelUCB seems to be limited.
>
> **[R2]** We want to clarify that DisKernelUCB is mainly used to demonstrate that a direct kernelization of the prior distributed linear bandit algorithm, DisLinUCB, does not work, due to its required transfer of raw data samples (mentioned in the beginning of Section 3.2). This motivates us to use Nyström approximation in our main algorithm Approx-DisKernelUCB, so that the clients only need to transfer the approximated statistics of size $\gamma_{NT} \times \gamma_{NT}$, which guarantees sub-linear communication cost.
>
> That being said, the analysis of DisKernelUCB requires special attention when constructing the filtration and applying the self-normalized bound as discussed in Appendix B, which is not a simple extension of the centralized algorithm. These technical results serve as the foundation for the analysis of our main algorithm Approx-DisKernelUCB.
>
>
> **[Q3]** What is the relations between the regret bound of Approx-DisKernelUCB and the Nystrom method to map x? How the approximation affect the regret bound? I think it is useful if authors can elaborate more on this in the paper.
>
> **[R3]** The regret bound of Approx-DisKernelUCB increases with respect to the approximation error of the Nystrom method, which is denoted as $\epsilon$ (see line 718 in appendix). As shown in Theorem 4.4, our result holds true for any constant value of $\epsilon$ that is smaller than $\frac{1}{3}$. We have added some explanation about this in line 258 in the updated version.
>
>
> **[Q4]** The paper assumes that the rewards are generated using a kernel function belongs to an RKHS.
>
> **[R4]** We want to emphasize that all prior works in distributed bandits assume context-free [24] or contextual linear models [19, 30], and our extension to functions in RKHS is already a strict generalization of all these existing works. Moreover, we have demonstrated in our empirical evaluations on real-world datasets that the proposed algorithm consistently outperforms prior works that use linear models.

---

> > ### Comment · Reviewer_fUXs · 2022-08-08
> > **Discussion**
> >
> > Thanks the authors for responding to my comments. The authors' response could not address most of my main concerns. I still believe that this work lacks preserving the privacy of users. Even if users trusts each other sending data samples to the server (either all or a portion of it) put the privacy of users at risk. I believe the proposed algorithms in the paper are comparable with federated learning since both approaches enable collaboration of users with each other using a central server. However, authors designed the collaboration of users in such a way that users need to send their data samples to the server. On the other hand, in federated learning users do not need to send their data samples to the server while they are collaborating. This shows the advantage of federated learning based methods to the proposed algorithms. In addition, I think the assumption that rewards are generated using a kernel function belongs to an RKHS may be unlikely to be the case in realistic scenarios. Can authors give some realistic examples? In such cases, one can ignore the context and use the context-free multi-armed bandit federated learning which can preserve the privacy. To sum up, I believe that the proposed algorithms suffer from lack of privacy preservation and it is necessary for this paper to compare the proposed algorithms both theoretically and empirically with existing federated learning counterparts.

---

> > > ### Author Response · Authors · 2022-08-08
> > > **Response to Reviewer fUXs [Part 3/3]**
> > >
> > > References
> > >
> > > [A1] Srinivas, N., Krause, A., Kakade, S.M. and Seeger, M., 2009. Gaussian process optimization in the bandit setting: No regret and experimental design. arXiv preprint arXiv:0912.3995.
> > >
> > > [A2] Shahriari, B., Swersky, K., Wang, Z., Adams, R.P. and De Freitas, N., 2015. Taking the human out of the loop: A review of Bayesian optimization. Proceedings of the IEEE, 104(1), pp.148-175.
> > >
> > > [A3] Snoek, J., Larochelle, H. and Adams, R.P., 2012. Practical bayesian optimization of machine learning algorithms. Advances in neural information processing systems, 25.
> > >
> > > [A4] Kandasamy, K., Neiswanger, W., Schneider, J., Poczos, B. and Xing, E.P., 2018. Neural architecture search with bayesian optimisation and optimal transport. Advances in neural information processing systems, 31.
> > >
> > > [A5] Bekkerman, R., Bilenko, M. and Langford, J. eds., 2011. Scaling up machine learning: Parallel and distributed approaches. Cambridge University Press.
> > >
> > > [A6] Molzahn, D.K., Dörfler, F., Sandberg, H., Low, S.H., Chakrabarti, S., Baldick, R. and Lavaei, J., 2017. A survey of distributed optimization and control algorithms for electric power systems. IEEE Transactions on Smart Grid, 8(6), pp.2941-2962.
> > >
> > > [A7] Rabbat, M. and Nowak, R., 2004, April. Distributed optimization in sensor networks. In Proceedings of the 3rd international symposium on Information processing in sensor networks (pp. 20-27).
> > >
> > > [A8] Lee, H., Lee, S.H. and Quek, T.Q., 2019. Deep learning for distributed optimization: Applications to wireless resource management. IEEE Journal on Selected Areas in Communications, 37(10), pp.2251-2266.
> > >
> > > [A9] Shariff, R. and Sheffet, O., 2018. Differentially private contextual linear bandits. Advances in Neural Information Processing Systems, 31.
> > >
> > > [A10] Neel, S. and Roth, A., 2018, July. Mitigating bias in adaptive data gathering via differential privacy. In International Conference on Machine Learning (pp. 3720-3729). PMLR.
> > >
> > > [A11] McMahan, B., Moore, E., Ramage, D., Hampson, S. and y Arcas, B.A., 2017, April. Communication-efficient learning of deep networks from decentralized data. In Artificial intelligence and statistics (pp. 1273-1282). PMLR.
> > >
> > > [A12] Karimireddy, S.P., Kale, S., Mohri, M., Reddi, S., Stich, S. and Suresh, A.T., 2020, November. Scaffold: Stochastic controlled averaging for federated learning. In International Conference on Machine Learning (pp. 5132-5143). PMLR.
> > >
> > > [A13] Mitra, A., Jaafar, R., Pappas, G.J. and Hassani, H., 2021. Linear convergence in federated learning: Tackling client heterogeneity and sparse gradients. Advances in Neural Information Processing Systems, 34, pp.14606-14619.
> > >
> > > [A14] Kairouz, P., McMahan, H.B., Avent, B., Bellet, A., Bennis, M., Bhagoji, A.N., Bonawitz, K., Charles, Z., Cormode, G., Cummings, R. and D’Oliveira, R.G., 2021. Advances and open problems in federated learning. Foundations and Trends® in Machine Learning, 14(1–2), pp.1-210.
> > >
> > > [A15] Geyer, R.C., Klein, T. and Nabi, M., 2017. Differentially private federated learning: A client level perspective. arXiv preprint arXiv:1712.07557.
> > >
> > > [A16] Mothukuri, V., Parizi, R.M., Pouriyeh, S., Huang, Y., Dehghantanha, A. and Srivastava, G., 2021. A survey on security and privacy of federated learning. Future Generation Computer Systems, 115, pp.619-640.

---

> > > ### Author Response · Authors · 2022-08-08
> > > **Response to Reviewer fUXs [Part 2/3]**
> > >
> > > **[Q3]** “I believe the proposed algorithms in the paper are comparable with federated learning since both approaches enable collaboration of users with each other using a central server. However, authors designed the collaboration of users in such a way that users need to send their data samples to the server. On the other hand, in federated learning users do not need to send their data samples to the server while they are collaborating. This shows the advantage of federated learning based methods to the proposed algorithms.”
> > >
> > > **[R3]**
> > > We need to articulate that the problem studied in this paper is **distributed machine learning** [A5], rather than federated learning. Federated learning is a special case of distributed machine learning with additional restrictions, e.g., the transfer of data samples is prohibited, which nowadays attracts increasing attention in applications like distributed recommender systems. However, distributed machine learning, as a much broader scope, also has a vast spectrum of application scenarios in practice, e.g. power systems [A6], sensor networks [A7], wireless communication [A8], etc., and therefore should not be ignored.
> > >
> > > For example, as we mentioned in our response [R1], an important application of kernel bandit / Gaussian process optimization is experimental design and AutoML; and in a distributed setting, the clients there correspond to computing machineries belonging to the same entity, which is different from the typical application scenarios of federated learning, where the clients are mobile phones belonging to different users / data centers (i.e., communication happens across different entities).
> > >
> > > In such cases, privacy is not the concern, but instead, more attention should be put on whether the adopted learning model can well fit the dataset encountered in various scenarios. This is exactly the strength of kernel methods, as it offers a wide range of non-linear models to choose from. And the main contribution of our paper is to propose the first distributed bandit algorithm that takes advantage of such strength of kernel methods, while still being communication efficient. Hence, we highly suggest our readers not to narrowly understand the notion of clients in our paper as individual users (such as those in a recommender system), but general autonomous agents seeking to collaborate with others for improved learning utility.
> > >
> > > Moreover, we cannot agree that an algorithm that does not send raw data automatically achieves privacy. As discussed in our previous response, existing federated contextual bandits, including the federated MAB suggested by the reviewer in Q2, are not private under specific formal privacy notions. For example, in DisLinUCB, the sufficient statistics are directly shared with the server and across agents. Under the notion of differential privacy, neither the reward nor the arms are kept private [A9, A10]. As federated MAB shares reward and number of arm pulls, it is not private either. And we should point out that even those standard federated learning algorithms (such as FedAvg [A11], SCAFFOLD [A12], and FedLin [A13]) that only share gradients/models are not automatically private, which has been extensively discussed in literature (e.g., Section 4 of [A14]). Depending on the required privacy definition, specific treatments are needed, e.g. randomized mechanisms to ensure differential privacy of federated learning [A15, A16]. Hence, we believe a completely private federated bandit solution needs much more thorough exploration and is beyond the scope of this paper, which is distributed kernel bandit. But we would like to have this as a future extension of our work.

---

> > > ### Author Response · Authors · 2022-08-08
> > > **Response to Reviewer fUXs [Part 1/3]**
> > >
> > > **[Q1]** I think the assumption that rewards are generated using a kernel function belongs to an RKHS may be unlikely to be the case in realistic scenarios.
> > >
> > > **[R1]**
> > > The so-called reward function in bandit problems is basically a mapping from input observations about an arm, aka, the context, to its observed reward. As in general machine learning literature, the kernel method is a generalization of explicit parametric methods; for example, the linear kernel, i.e., setting $k(x, x^{\prime})=x^{\top} x^{\prime}$ in line 116 of our paper, recovers the linear function mapping. It introduces **great flexibility in model choices**, and covers a wide range of nonlinear mappings, e.g., by choosing Gaussian kernel, Sigmoid kernel, Polynomial kernel, etc., as the kernel function $k(\cdot, \cdot)$ without the need of explicitly defining the mapping function. More importantly, the corresponding **analysis is agnostic to the specific choice of kernel**. Hence, kernel bandit [6, 29] is nothing but bandit learning taking advantage of kernel functions to handle possible non-linear reward mappings and its generality in regret analysis. This is exactly what we are working on in this paper, as our analysis is not restricted to any specific kernel functions.
> > >
> > > Another possible interpretation of the reviewer’s argument here could be that the reviewer believes the functional space defined by RKHS is too restrictive (or small) to cover realistic scenarios. This however we cannot agree on, as RKHS is of particular importance to the field of statistical learning theory, and many popularly used functions belong to this space. If a kernelized reward mapping is considered as unrealistic, is it equivalent to considering kernel methods unrealistic for machine learning?
> > >
> > > As we try to understand the reviewer’s comment, if the reviewer believes the reward assumptions made in prior works of federated bandit learning are realistic, then as we mentioned in our previous response, our setting can be easily specialized to **contextual-linear setting** [14, 19] (by using a linear kernel), and **context-free setting** [30] (e.g. by setting $k(x, x^{\prime})=1$ if $x = x^{\prime}$ and $k(x, x^{\prime})=0$ otherwise), considered in these prior works. Apart from that, the **neural tangent kernel** studied in recent works of neural bandit under centralized setting [32] is also covered by this assumption. Moreover, the vast amount of works in **Gaussian process optimization**, which are popularly used for experimental design [A1, A2], AutoML [A3, A4], etc., are also special cases of ours, i.e., kernelized bandits with Gaussian kernels.
> > >
> > > Therefore, we argue that the kernel function is already very general and thus a much weaker assumption about the reward mapping compared with prior works. Besides, it has also demonstrated superior performance in the application scenarios mentioned above. Maybe the reviewer wants to provide some example function assumptions to elaborate what a more realistic reward function means here, so that we can better address your concern.
> > >
> > >  **[Q2]** In such cases, one can ignore the context and use the context-free multi-armed bandit federated learning which can preserve the privacy.
> > >
> > > **[R2]**
> > > We had difficulties in understanding the reviewer’s comment here. As it is commonly known that context-free MAB ignores the correlation among arms, i.e., observations from one arm can bring additional knowledge about other arms, which is critically important for faster convergence and lower regret in contextual bandits. Or more specifically, the regret of context-free MAB scales w.r.t. the number of arms, while the regret of contextual bandits scales w.r.t. the context dimension that is usually much smaller. This is also why context-free MAB only works with a fixed arm set, and the community has spent so much effort in contextual bandits, including kernel bandits [6, 29]. We could not understand why anyone would ignore the already available context to receive deteriorated utility.
> > >
> > > However, on the other hand, if we restrict ourselves to a fixed arm set as in federated MAB (i.e., set $A_{t}=A$ in line 107), our kernel bandit solution also does not need to transfer raw data, but instead only the mean rewards and number of pulls of each arm, which is exactly the same as prior works in federated MAB [24, 30]. Moreover, we can utilize the available context features for improved regret in this situation without sending raw data.
> > >
> > > Again, we need to emphasize that the kernel bandit should be understood as a family of bandit algorithms, which can be realized by specific choices of kernel functions for different applications. And therefore our solution is general and agnostic to the chosen kernel.

---

### Official Review · Reviewer_nxdg · 2022-07-10

**Rating:** 6
**Confidence:** 4
**Soundness:** 3 good
**Presentation:** 3 good
**Contribution:** 4 excellent

**Summary:**

This paper investigates the communication efficiency challenge of kernelized contextual bandits in a distributed setting. This paper resolves this challenge by equipping all agents to communicate via a common Nyström embedding which is updated adaptively as more data points are collected. This paper proposes an algorithm named Approx-DisKernelUCB and demonstrated that Approx-DisKernelUCB can achieve sub-linear rate in both regret and communication cost.

**Questions:**

Could you explain the differences of the techniques used to handle communication challenges between this work and prior distributed linear bandit works?

Can the Nyström embedding technique solve the communication challenges of distributed linear bandits?

Why not directly present algorithm Approx-DisKernelUCB? The algorithm DisKernelUCB seems not satisfactory due to its linear communication cost.


**Limitations:**

See the weakness above.

**Strengths And Weaknesses:**

**Strengths:**
The proposed Nyström embedding technique is very interesting, which makes a solid contribution to the communication challenge of distributed kernelized contextual bandits. The proposed algorithm Approx-DisKernelUCB achieves good performance in terms of both regret and communication costs.

This paper is well-written and the presented theoretical analysis looks sound and solid. Readers can easily understand the main idea behind algorithm Approx-DisKernelUCB.

**Weaknesses:**

The structure of this paper could be further improved. I suggest the authors to give more intuition behind the theoretical analysis and put some representative experimental results in the main text.

It would be better if the authors can comment and discuss the following related works on distributed pure exploration in the literature review part.

1. Distributed exploration in multi-armed bandits. Hillel et al.
2. Collaborative learning with limited interaction: Tight bounds for distributed exploration in multi-armed bandits. Tao et al.
3. Collaborative Pure Exploration in Kernel Bandit. Du et al.

---

> ### Author Response · Authors · 2022-08-02
> **Response to Reviewer nxdg**
>
> We have improved our presentation of the theoretical analysis, moved the experiment results in the main paper, and included the suggested references in the revised version.
>
> **[Q1]** Could you explain the differences of the techniques used to handle communication challenges between this work and prior distributed linear bandit works?
>
> **[R1]** We have demonstrated via the analysis of the baseline algorithm DisKernelUCB (Algorithm 1) that a direct kernelization of the prior distributed linear bandit algorithm, DisLinUCB, does not work, because even though the total number of global synchronizations is reduced, the transfer of raw data still leads to linear communication cost as discussed in Remark 1. Compared with the distributed linear bandits, where the clients only need to transfer the sufficient statistics of size $d \times d$ for global model update, this required transfer of raw data samples in distributed kernelized bandits posits a key challenge in improving its communication efficiency [9] (this is the index of citations in the updated version of our paper). Therefore, we proposed to use Nyström approximation, so that the clients only need to transfer the approximated statistics of size $\gamma_{NT} \times \gamma_{NT}$, which guarantees sub-linear communication cost.
>
> However, since each client now only has access to the approximated statistics, the event-trigger based on the exact determinant ratio in Eq (2) (kernelized version of the event-trigger used in DisLinUCB) no longer applies. Instead, we have to resort to the one based on the approximated variances in Eq (4), which requires the unique proof techniques in the regret and communication cost analysis as mentioned in line 670 in Appendix G.1.
>
>
> **[Q2]** Can the Nyström embedding technique solve the communication challenges of distributed linear bandits?
>
> **[R2]** Since distributed linear bandits are just distributed kernelized bandits with a linear kernel function, our proposed algorithm based on the Nyström approximation can be directly applied. However, we should note that, the main motivation of using the kernelized/dual form of the estimator (line 149) instead of the primal form (line 146) is to avoid dependence on $p$, i.e., the dimension of RKHS that is possibly infinite. For distributed linear bandits, since $p=d$, it is more communication-efficient to directly work under the primal form. For example, the existing distributed linear bandit algorithms like DisLinUCB, directly transfers the $d \times d$ statistics with no need of approximation.
>
> To avoid our possible misunderstanding of the reviewer’s question, we need to confirm if the reviewer is asking whether our proposed algorithm can address the communication challenge of distributed linear bandits *when $d$ is very large*, then the answer is negative. As shown in Theorem 3.3., the communication cost of Approx-DisKernelUCB has an $O(\gamma_{NT}^{3})$ scaling, and for linear kernel, $\gamma_{NT} = d \log(NT)$. As a result, it has the same $O(d^{3})$ scaling as DisLinUCB. To further reduce the scaling in $d$, one can apply matrix sketching methods on the A matrix in the primal form (line 146) as [A2], which helps reduce the scaling to $O(d^{2})$.
>
> [A2] Kuzborskij, I., Cella, L. and Cesa-Bianchi, N., 2019, April. Efficient linear bandits through matrix sketching. In The 22nd International Conference on Artificial Intelligence and Statistics (pp. 177-185). PMLR.
>
>
> **[Q3]** Why not directly present algorithm Approx-DisKernelUCB? The algorithm DisKernelUCB seems not satisfactory due to its linear communication cost.
>
> **[R3]** As mentioned in the beginning of Section 3.2, DisKernelUCB is used to demonstrate that a direct kernelization of the prior distributed linear bandit algorithm, DisLinUCB, will not work due to its required transfer of raw data samples. This motivates us to use the Nyström approximation method in Approx-DisKernelUCB, so that the clients only need to transfer the approximated statistics of size $\gamma_{NT} \times \gamma_{NT}$, which guarantees sub-linear communication cost.
>
> Moreover, the technical results obtained in the derivation of the confidence ellipsoid (Appendix B) and the analysis for DisKernelUCB (Appendix C) serve as the foundation for the analysis of Approx-DisKernelUCB.

---

> > ### Comment · Reviewer_nxdg · 2022-08-09
> > **Response to Authors**
> >
> > Thank you very much for your clear reply. My concerns were addressed. I understand that the main focus of this paper is to improve the dependence of the communication cost from the high dimension $p$ to the maximum information gain $\gamma_{NT}$.
> > I suggest the authors to include more discussion and comparison with existing distributed/federated linear/kernel bandit works to faciliate the readers' understanding of your contributions on communication.
> >
> > I keep my score.

---

### Official Review · Reviewer_h26a · 2022-07-17

**Rating:** 6
**Confidence:** 3
**Soundness:** 3 good
**Presentation:** 3 good
**Contribution:** 2 fair

**Summary:**

This paper studies the distributed kernel bandit problems in a decentralized setting. As a first observation, they consider Distributed-Kernel UCB, which is naturally derived from the one for distributed linear bandits, and prove that its communication cost suffers from the linear dependence of time horizon T. To remedy this, they propose Approximated Distributed Kernel UCB that uses the Nyström approximation techniques. The resulting communication depends on the term on maximum information gain $\gamma_{NT}^3$, removing the linear dependence of T. They also conducted a series of experiments using real-world dataset.


**Questions:**

Some may say that the key techniques are essentially imported from the work of [2,3] and combining these with distributed Kernel bandits seems to be a straightforward approach. It would be appreciated if the authors explain what difficulties arise when applying them to distributed kernel bandits.

**Limitations:**

-

**Strengths And Weaknesses:**

Strength:
The idea of Nyström approximation techniques enables the improved Dist Kernel UCB to have a better communication cost than the naive Kernel UCB. Approximated Kernel UCB maintains the good representation of the whole dataset, which is used to escape the sending of all unshared raw data.
In order to deal with such an approximated estimator, the event-trigger in Eq (4) is derived for determining when to incur the global synchronization.

The size of the dictionary $S_t$ scales linearly w.r.t. the maximum information gain $\gamma_{NT}$, which therefore improves the communication efficiency.
All of the parameters such as threshold D in the event-trigger and how the subroutine of RLS in algorithm 3 is invoked is clearly stated.

The value of the information gain $\gamma_{NT}$ is now a key factor in the analysis. They satisfactorily discussed what this factor implies and which affected the factor, when it is sufficiently smaller than $T^{1/3}$.


Weakness:
The lower bound on communication costs for this problem is not fully discussed.

===Post Rebuttal===
Thank you for author's response and detailed discussion for communication lower bound result with Square Exponential kernels.
I also read other reviews and keep my score for acceptance.

---

> ### Author Response · Authors · 2022-08-02
> **Response to Reviewer h26a**
>
> **[Q1]** The lower bound on communication costs for this problem is not fully discussed
>
> **[R1]** To facilitate discussion about the communication lower bound, we should note that, for distributed bandit problems, the communication lower bound is only meaningful under different constraints/requirements on regret. The following two extreme cases of regret are of particular interests to explain the notion:
> 1. Run $N$ instances of an optimal bandit algorithm, e.g., LinUCB for linear bandit and KernelUCB for kernelized bandit, separately on each client with no communication, which leads to $O(N\sqrt{T})$ regret and $0$ communication cost;
> 2. Run one instance of the optimal bandit algorithm over the data of all $N$ clients, which leads to $O(\sqrt{NT})$ regret and communication cost linear in $NT$.
> The regret upper bound in **case 2** is optimal, since it already matches the lower bound for a centralized bandit problem of time horizon $NT$. Therefore, the goal of distributed bandit algorithms is to attain the optimal regret of $O(\sqrt{NT})$, while having a communication cost sub-linear in $T$.
>
> The main contribution of our paper is to propose the first algorithm that achieves such a goal for the distributed kernelized bandit problem. The lower bound analysis for the communication cost of distributed bandits is highly non-trivial, and still remains an open problem. We leave this as an important future direction of our work.
>
> To the best of our knowledge, the only communication lower bound result available is for **case 1**. It was originally provided for the context-free setting (Theorem 2 of [30]), but as shown in a recent work (Theorem 5.3 of [A1]), it also applies to the linear setting. It states that, in order to have smaller regret than $O(N\sqrt{T})$ in **case 1**, an $\Omega(N)$ **number of communications** is necessary.
>
> For the sake of completeness, we have also added a communication lower bound result for the distributed kernelized bandits with **Square Exponential kernels** in the revised version of our paper (see Appendix I), following the same proof procedure.
>
> So far, there is no useful result for **case 2**, i.e., the communication lower bound to obtain $O(\sqrt{NT})$ regret; and none of the existing distributed linear bandit algorithms [19, 30, A1] can close the gap with the $\Omega(N)$ lower bound for **case 1**. Specifically, to obtain $O(\sqrt{NT})$ regret, DisLinUCB [30] requires $O(d N^{1.5}\log(NT))$ number of communications,  AsyncLinUCB [19] and FedLinUCB [A1] require $O(d N^{2}\log(NT))$ number of communications, with the latter two having the same scaling in $N$ as our Approx-DisKernelUCB (which requires $O(N^{2}\gamma_{NT})$ number of communications).
>
> [A1] He, J., Wang, T., Min, Y. and Gu, Q., 2022. A Simple and Provably Efficient Algorithm for Asynchronous Federated Contextual Linear Bandits. arXiv preprint arXiv:2207.03106.
>
> **[Q2]**  Some may say that the key techniques are essentially imported from the work of [2,3] and combining these with distributed Kernel bandits seems to be a straightforward approach. It would be appreciated if the authors explain what difficulties arise when applying them to distributed kernel bandits.
>
> **[R2]**  The Nyström approximation method was first adopted in [3, 4] (these are the new indices for the mentioned references in the updated version of our paper) to reduce computational complexity of Gaussian process optimization, but our paper is the first to adopt it in the distributed kernelized bandit setting to **reduce communication cost**. This difference in the problem setting introduces several new challenges compared with [3, 4].
>
> First, as mentioned in line 459 in Appendix B (of our updated paper), in the decentralized bandit setting, special care needs to be taken when constructing the filtration, which is different from the centralized setting considered in [3, 4]. We proposed a modified version of the standard self-normalized bound in Lemma B.3 to address this issue in a rigorous manner.
> Second, in our setting the event-trigger can only be computed based on each client’s local data, instead of the centralized data as in [4]. This introduces extra delays in the triggering of the global synchronizations, which requires different procedures to bound the regret as shown in Appendix G.1 and Appendix G.2.
> Moreover, as mentioned in line 262, the delay in triggering global synchronization also affects the quality of the dictionary of Approx-DisKernelUCB, and thus the approximation error depends on the threshold $D$, instead of being a constant as in [4]. This leads to Lemma 4.1, which is unique to our solution.

---

### Official Review · Reviewer_tP3Z · 2022-08-29

**Rating:** 6
**Confidence:** 3
**Soundness:** 3 good
**Presentation:** 3 good
**Contribution:** 4 excellent

**Summary:**

The current paper studies the distributed kernelized bandit, a generalization of the distributed linear bandit. Different from the distributed linear bandit [25], one cannot upload/download sufficient statistics (i.e., Gram matrix) for total $O(\log T)$ communication rounds, since for later communication rounds, sending either the raw data (size of $dT$ to resume the sufficient statistics) or the original kernelized Gram matrix (size of $p^2$, where p can be infinite) will be prohibitively expensive. The author adopts the idea of sparse approximation, i.e., Nyström embedding, to project the $p\times p$ kernelized Gram matrix into a new matrix that is of size $O(\gamma_{NT})$, where $O(\gamma_{NT}$ is the information gain of order $O(\log T)$ for some kernel functions. To this end, the authors can prove $O(\sqrt{NT}\gamma_{NT})$ regret and $O(N^2\gamma_{NT}^3)$ communication cost.

**Questions:**

Can you discuss and explain more about the tightness of your result? This could be comparing with the degenerate case, the lower bound, or the trade-off between the communication cost and the regret.

**Limitations:**

The authors adequately addressed the limitations and there is no foreseen potential negative societal impact of the current work

**Strengths And Weaknesses:**

Strength.

1. The problem is well-motivated. To the best of my knowledge, there are no previous works studying distributed kernelized bandit, with rigorous sublinear regret and sublinear communication cost guarantee. I enjoy the way how the authors first discuss the straightforward usage of the previous solution [25] as a warm-up and motivate the challenge of the problem.

2. From the technical point of view, though I did not go through all the proofs, the intuition of the current paper is clear and reasonable. I think the authors borrow the batched/delayed kernelized bandit as building blocks to save the communication round and use the sparse approximation to save the per-round data to be uploaded/downloaded. The main contributions of this paper are designing a suitable triggering event with only approximated sufficient statistics at hand and using ridge leverage score sampling to send sampled information. From my perspective, the solution and the analysis should be of independent interest to other related works.

Weakness.

1. Unlike [25], I do not find any discussion about the tightness of the result and the communication and regret trade-off. Also, there are no lower bound results for communication cost, so it is a little bit confusing how tight the communication cost $O(N^2\gamma_{NT}^3)$ is. Moreover, it is not clear to me why there are no $d$ factors in the regret, I guess the reason is you replace both $d$ and $T$ to $\gamma_{NT}$ so you can replace the trivial bound (for Algorithm 1) to $O(N^2\gamma_{NT}^2)$, but this again is one $O(\gamma_{NT})$ factor less than your current result.

---

### Meta-Review · Area_Chair_CcKz · 2022-08-30

**Recommendation:** Accept
**Confidence:** Certain

**Metareview:**

The paper studies distributed kernelized bandit and applies the Nyström approximation to achieve communication efficiency, which is a novel technique appreciated by the reviewers. After some discussions, and including adding another extra reviewer, I believe the paper has enough contribution and is worth to be published at NeurIPS'2022. However, the authors do need to take all review comments serious and give a thorough revision to the paper, including comparisons to other possible approaches, such as federated learning. The authors also need to make it very clear why studying kernelized bandit and distributed bandit setting is important for applications.

**Award:**

No

---

### Decision · Program_Chairs · 2022-09-14

Accept